# Citizen observations contributing to flood modelling: opportunities and challenges

Thaine Herman Assumpção[1], Ioana Popescu[1], Andreja Jonoski[1], Dimitri P. Solomatine[1, 2]

[1]Integrated Water Systems and Governance, IHE Delft, Delft, The Netherlands
[2]Water Resources Section, Delft University of Technology, Delft, The Netherlands

*Correspondence to*: Thaine Herman Assumpção (t.hermanassumpcao@un-ihe.org)

**Abstract.** Citizen contributions to science have been successfully implemented in many fields – and water resources is one of them. Through citizens, it is possible to collect data and obtain a more integrated decision-making process. Specifically, data scarcity has always been an issue in flood modelling, which has been addressed in the last decades by remote sensing and is already being discussed in a citizen science scenario. In this context, this article aims to review the literature on the topic and analyse the opportunities and challenges that lie ahead. The literature on monitoring, mapping and modelling, was evaluated according to the flood-related variable citizens contributed to. Pros and cons of the collection/analysis methods were summarised. Then, pertinent publications were mapped into the flood modelling cycle, considering how citizen data properties (spatial and temporal coverage, uncertainty and volume) are related to its integration into modelling. It was clear that the number of studies in the area is rising. There are positive experiences reported in collection and analysis methods, for instance with velocity and land cover, and also when modelling is concerned, for example by using social media mining. However, matching the data properties necessary for each part of the modelling cycle with citizen generated data is still challenging. Nevertheless, the concept that citizen contributions can be used for simulation and forecasting is proved and further work lies in continuing developing and improving not only methods for collection and analysis but certainly for integration into models as well. Finally, in view of recent automated sensors and satellite technologies, it is through studies as the ones analysed in this article that the value of citizen contributions is demonstrated, as complement to such technologies.

## 1 Introduction

The necessity to understand and predict the behaviour of floods has been present in societies around the world. This comes from the fact that floods impact their surroundings - in negative or in positive ways. The most common way used nowadays to better understand and often predict flood behaviour is through modelling and, depending on the system at hand, a variety of models can be used (Teng et al., 2017).

In order to have adequate representation of floods, most models require large amounts of data, both for model building and model usage. This is especially true for pluvial flood modelling, where flooding may not occur in gauged rivers and hence, flow gauging stations outside of flooded zones may be of little use. Remote sensing technologies are a part of the solution, as they offer spatially distributed information. However, their availability may be limited, also in terms of space and time, and their uncertainties often are not quantifiable (Di Baldassarre et al., 2011; Grimaldi et al., 2016; Jiang et al., 2014; Li et al., 2017). Thus, acquiring the necessary data for simulations and predictions can still be expensive, particularly for rapidly changing systems that require frequent model updates.

In this context, sources of data coming in abundance and at low-costs are needed, together with modified modelling approaches that can use these data and can adapt to changes as fast as they occur. Citizen Observatory (CO) is an emerging concept in which citizens monitor the environment around them (Montargil and Santos, 2017). It is often considered under the umbrella of Citizen Science (including citizen participation up to the scientist level) and it is also related to the concept of crowdsourcing (distributing a task among many agents). With technology at hand, it is possible to empower citizens to not only participate in the acquisition of data but also in the process of scientific analysis and even in the consequent decision-making process (Evers et al., 2016). Citizen Observatories have been researched in several EU-funded projects. Finished projects (CITI-SENSE, Citclops, COBWEB, OMNISCIENTIS and WeSenseIt) already resulted in valuable contributions to the field (Alfonso et al., 2015; Aspuru et al., 2016; Friedrichs et al., 2014; Higgins et al., 2016; Uhrner et al., 2013). For example, the CITI-SENSE project managed to simultaneously collect perception data and acoustic measurements in an approach that can be used to develop citizen empowerment initiatives in case of noise management (Aspuru et al. 2016); while in COBWEB project processes of quality assurance, data conflation and data fusion were studied and recommendations were made (Friedrichs et al., 2014). The currently running CO projects (Ground Truth 2.0, LANDSENSE, SCENT and GROW Observatory) propose to investigate this concept further.

Citizen science concepts have been researched and applied in various fields such as ecology and galaxy inspection (Lintott et al., 2008; Miller-Rushing et al., 2012). Volunteer Geographic Information (VGI), as one of the most active citizen science areas, has developed over the past decade and several researchers reviewed the state of the art of citizen science in the field of geosciences (Heipke, 2010; Klonner et al., 2016). There is also a part of the scientific community dedicated to investigating damage data crowdsourced after flood emergencies (Dashti et al., 2014; Oxendine et al., 2014) and evaluating the cycle of disaster management (Horita et al., 2013). In the context of water resources, Buytaert et al. (2014) reviewed and discussed the contribution of citizen science to hydrology and water resources, addressing the level of engagement, the type of data collected (e.g. precipitation, water level) and case studies where more participatory approaches are being implemented. Le Coz et al. (2016) provided examples and reflections from three projects related to flood hydrology and crowdsourcing, which involve the derivation of hydraulic information from pictures and videos in Argentina, France and New Zealand.

The present review aims to look at studies that had citizen science connected to floods. Specifically, it focusses on the data collected by citizens that are relevant in a flood modelling context, benchmarking difficulties and benefits of their collection and integration into models. Integration is considered for the purposes of model set up, calibration, validation, simulation and forecasting.

The review process involved defining web platforms, keywords and criteria for searching and selecting publications. The main platforms used were Scopus and Google Scholar. The keywords are a combination of words related to citizen science (e.g. "citizen science" and crowdsourcing) and to flood-related variables (e.g. "water level" and "flood extent"). The obtained articles were scanned for their content. Articles were selected mainly if crowdsourced data was obtained for quantitative use in monitoring, mapping or modelling. There were studies that were not selected because they just mention the use of crowdsourced data and do not provide more relevant information on collection, analysis, use and quantity of data, such as Merkuryeva et al. (2015). The same is the case of studies that evaluate variables qualitatively, in ways that cannot be directly associated with modelling (Kim et al., 2011). This review included articles published up to April 2017.

Further in this section, we introduce the citizen science concept and related classification systems. In Sect. 2 of the article, we overview studies on citizen contributions for flood modelling, classifying them according to the flood-related variable the contributions were made, followed by a summary of the pros and cons of measurement and analysis methods. Section 3 aggregates the studies that involve flood modelling and analyses the contributions considering the component of the modelling process where they were used, also including a discussion of the factors that affect flood modelling. Section 4 describes the challenges and opportunities of using data contributed by citizens in flood modelling, and finally, Sect. 5 presents the conclusions and recommendations.

## 1.1 Citizen Science

Buytaert et al. (2014) defined citizen science as "the participation of the general public (i.e. non-scientists) in the generation of new knowledge". In the same manner that the involvement of citizens can be diverse, such is the way their participation is found in the scientific literature:

- Citizen Science (Buytaert et al., 2014)
- Citizen Observatory (Degrossi et al., 2014)
- Citizen Sensing (Foody et al., 2013)
- Trained volunteers (Gallart et al., 2016)

- Participatory data collection methods (Michelsen et al., 2016)
- Crowdsourcing (Leibovici et al., 2015)
- Participatory sensing (Kotovirta et al., 2014)
- Community-based monitoring (Conrad and Hilchey, 2011)
- Volunteered Geographic Information (Klonner et al., 2016)
- Eye witnesses (Poser and Dransch, 2010)
- Non-authoritative sources (Schnebele et al., 2014)
- Human Sensor Network (Aulov et al., 2014)
- Crowdsourced Geographic Information (See et al., 2016)

Some of the terms used by the above-mentioned articles have specific definitions that are used to delineate debates on the social mechanisms of citizen participation. Others are just the best form the researcher found to characterise the contribution or the citizen (e.g. eye witnesses). Citizen Science and adjacent areas have become fields of research in themselves that, for instance, focus on understanding the motivation of citizens or its interaction with public institutions (Gharesifard and Wehn, 2016).

In this field, one of the classifications of citizen science is by level of engagement. Haklay (2013) built a model that has four levels (Fig. 1), in which the first one refers to the participation of citizens only as data collectors, passing through a second level in which citizens are asked to act as interpreters of data, going towards the participation in definition of the problem in the third level and finally, being fully involved in the scientific enterprise at hand. The review presented in this current article is focused on the contribution towards flood modelling only, coming most prominently from the two lowest levels of engagement. We do not discuss topics related to engagement for the generation of (quantitative) data. Further in this article, for readability, only the term crowdsourced data is used to refer to data from these two levels of engagement.

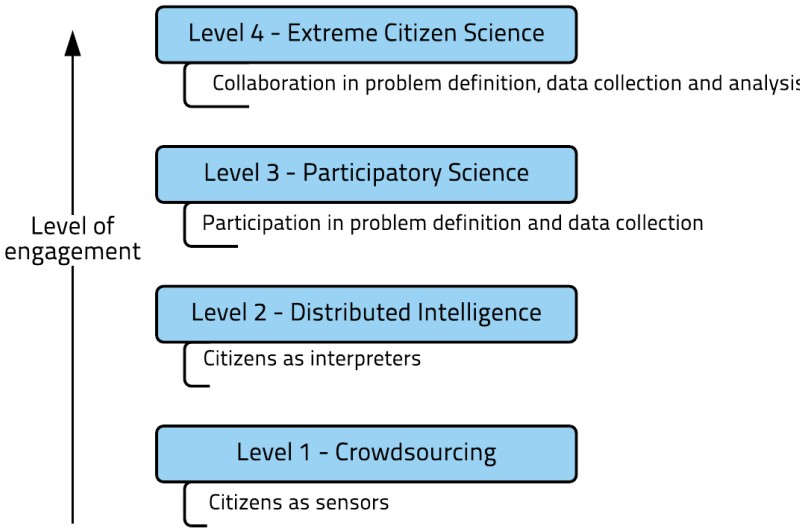

**Figure 1: Levels of participation and engagement in citizen science projects. Adapted from Haklay (2013).**

Another way to classify citizen science initiatives (within the context of VGI) is by setting them as implicitly/explicitly

5   volunteered and implicitly/explicitly geographic (Craglia et al., 2012). In this classification system, geographic refers to the main information conveyed through the contributed data, therefore, geo-tagged data is not necessarily geographic. For example, in the Degree Confluence Project (Iwao et al., 2006), citizens were oriented to go to certain locations, take pictures, make notes and deliberately make available their material on the project's website. In this case, the information is explicitly volunteered and geographic. Most land use/cover projects related to citizen science collect geographic information.

10  Differently, in the study conducted by Lowry and Fienen (2013) citizens would also willingly send text messages to the researchers, in this case providing water level readings from installed water level gauges. Although explicitly volunteered, the message was non-geographic (just geo-tagged). Another type of implicitly geographic information was derived from Twitter by Smith et al. (2015) to obtain flood water level, flow rate and flood inundation estimates. As the citizens did not make the information public with the specific purpose to provide estimates, it is implicitly volunteered.

The concepts defined by Craglia et al. (2012) can be graphically represented as in Fig. 2. The SCENT project[1] (Smart Toolbox for Engaging Citizens in a People-Centric Observation Web) is one of the four Horizon 2020-funded projects focussing on citizen observatories. It lies in the middle of this quadrant as it encourages citizens to participate in gaming to collect land cover/use data, in field campaigns to collect other implicitly geographic information (e.g. water level), and also

---

[1] https://scent-project.eu/

aims to obtain implicitly volunteered contributions through a CAPTCHA[2] plugin, in which citizens tag images, e.g. of land cover/use or water level, in order to access online content. Tagging images is uncorrelated to the CAPTCHA, it is a task performed after the test, on the same platform.

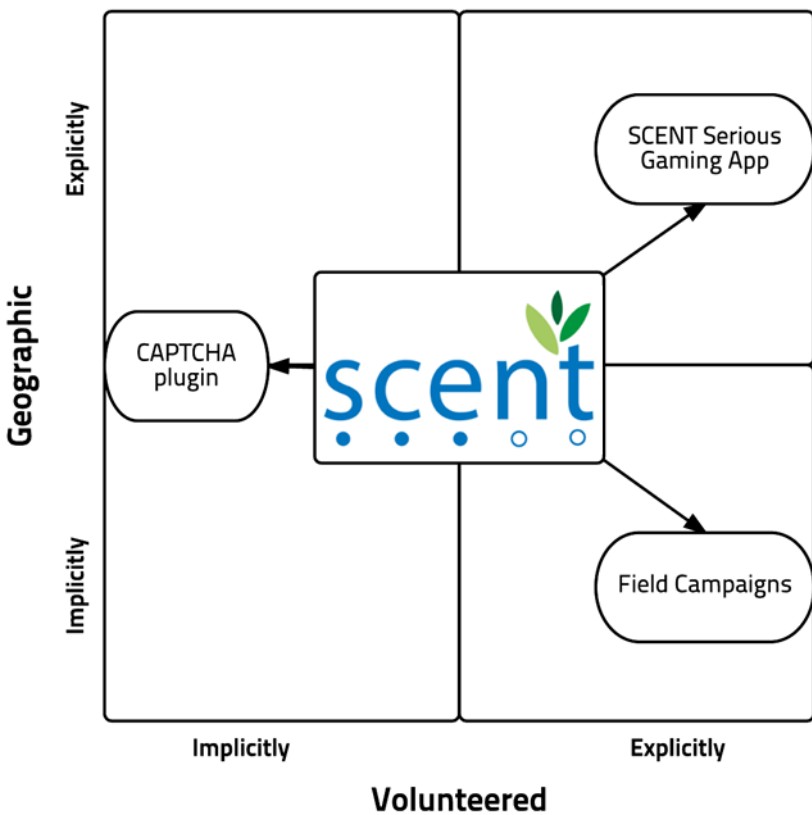

**Figure 2: SCENT project represented in the typology of VGI (Volunteered Geographic Information)**

## 2 Flood-related crowdsourced data

There are many types of data which relate to floods that can be collected by citizens. Likewise, there are many ways to collect, analyse and use them (for monitoring, mapping and modelling). In the next sub-sections we address how these aspects were explored in the scientific literature. Each sub-section discusses a data type corresponding to a flood modelling

---

[2] CAPTCHA stands for 'Completely Automated Public Turing test to tell Computers and Humans Apart'. It is a test evaluating if the subject is human, which is used in websites to provide security. After the test is done the user can be asked to perform extra tasks, for example, tag images.

variable: water level, velocity, flood extent, land cover and topography. Depending on the type of flooding, other variables are relevant, such as precipitation. The scientific literature already shows that citizens' contributions could be useful for observation this variable (Muller et al., 2015; De Vos et al., 2017). However, rainfall is not included in this section because it was already covered by the review of Muller et al. (2015). Moreover, in general it is a variable of greater importance for

hydrological models, whilst the present review is focussed on a hydrodynamic representation of floods[3]. Regarding the presented articles, there are some mentioned and reviewed in more than one section because they evaluated more than one variable, as it is, for example, the case of Smith et al. (2015).

## 2.1 Water level

Table 1 gives an overview of the articles about collection of water level data. The studies presented started to involve

citizens in the collection of water level data with the explicit goal of improving flood management. This is due to the ease of collecting such data, which mostly consists of comparing the water level with a clearly defined reference. In some cases, the reference is a water level gauge, the comparison is made by the citizen, and readings are being submitted to the researchers (Alfonso et al., 2010; Degrossi et al., 2014; Fava et al., 2014; Lowry and Fienen, 2013; Walker et al., 2016). Such kind of reading practically do not require further analysis, although they entail the installation of water level gauges.

In other cases, the citizen provides qualitative data that will be compared to references by researchers. Mostly during flooding situations, citizens provide pictures (Fohringer et al., 2015; Kutija et al., 2014; Li et al., 2017; McDougall, 2011; McDougall and Temple-Watts, 2012; Smith et al., 2015; Starkey et al., 2017) or videos (Le Boursicaud et al., 2016; Le Coz et al., 2016; Michelsen et al., 2016). In the case of pictures/images, the water level is compared with objects in the images

that have known or approximately known dimensions. For videos, although water level was estimated, the main goal was to obtain discharge values, via estimates of flow velocity. In two cases, texts from citizens were used (e.g. water over the knee), to calculate water level values or to assume a certain value when no value was provided (Li et al., 2017; Smith et al., 2015). This sort of data (text, pictures and videos) was mostly collected through social media and public image repositories. Gathering data from such sources requires mining of the relevant material (i.e. extraction of specific data from a dataset) and

dealing with uncertainties in the spatio-temporal characterization of the data of interest.

One aspect that varies across the studies is the level of detail in the comparison method used for determining the water level measurement. For example, McDougall (2011) and McDougall and Temple-Watts (2012) explicitly state that field visits to

---

[3] In general lines, hydrological models represent the transformation of rainfall into discharge, taking into account evapotranspiration and soil related processes, while hydrodynamic models represent the transformation of water levels and velocities in domains such as river reaches, taking into account ground roughness and topography. In both cases, there are many models, different in complexity and usage.

the selected photo locations are required in order to properly analyse the image and extract water level values. On the other hand, Fohringer et al. (2015), Smith et al. (2015) and Starkey et al. (2017) do not mention any method.

In most cases, crowdsourcing has been used to monitor water level, followed by the use of such data for modelling and lastly for mapping. In the case of Starkey et al. (2017), although hydrological modelling was done and water levels were converted into discharge to allow for comparisons, only qualitative comparisons were made.

**Table 1: Scientific literature on citizen contributions to measurement and analysis of water level**

| Study | Measurement/analysis methods | Type | Purpose | Flood type | Location |
|---|---|---|---|---|---|
| Alfonso et al. (2010) | Citizen's reading of water level gauges sent by text message | 1D | Monitoring | No flooding | The Netherlands |
| Lowry and Fienen (2013) | Citizen's reading of water level gauges sent by text message | 1D | Monitoring | No flooding | USA |
| Degrossi et al. (2014) | Citizen's reading of water level gauge sent through app/webpage | 1D | Monitoring | No flooding | Brazil |
| Walker et al. (2016) | Citizen's reading of water level gauge collected and provided by the community | 1D | Monitoring | No flooding | Ethiopia |
| Fava et al. (2014) | Citizen's reading of water level gauge sent through app/webpage | 1D | Modelling | Flood forecasting | Brazil |
| Le Boursicaud et al. (2016) | LSPIV analysis of video collected from social media (YouTube) | 1D | Monitoring | Flash flood | France |
| Le Coz et al. (2016) | LISPIV analysis of video sent through webpage | 2D | Modelling | Fluvial flood | Argentina |
| Michelsen et al. (2016) | Analysis of images extracted from videos collected from social media (YouTube) and own photographs | Neither | Monitoring | No flooding | Saudi Arabia |
| Li et al. (2017) | Analysis of texts and pictures collected from social media (Twitter) | 2D | Monitoring | Flood map | USA |
| Starkey et al. (2017) | Citizen's reading of water level gauge and analysis of pictures and videos collected from social media (Twitter) and crowdsourced (email, webpage and mobile app) | 2D | Monitoring | Flood | UK |
| McDougall (2011), McDougall and Temple-Watts (2012) | Analysis of texts and pictures collected from social media (Twitter, Facebook) and crowdsourced (email, text message, Ushahidi, Flickr and Picasa) | 2D | Mapping | Flood map | Australia |
| Kutija et al. (2014) | Analysis of pictures collected by the University and City Council | 2D | Modelling | Pluvial and drainage flood | UK |
| Aulov et al. (2014) | Visual analysis of texts and pictures collected from social media (Twitter and Instagram) | 2D | Modelling | Coastal flood | USA |
| Fohringer et al. (2015) | Visual analysis of pictures collected from social media (Twitter) and crowdsourced | 2D | Mapping | Flood | Germany |

| | (Flickr) | | | | |
| Smith et al. (2015) | Analysis of texts and pictures collected from social media (Twitter) | 2D | Modelling | Pluvial and drainage flood | UK |

## 2.2 Velocity

As velocities and discharges traditionally require more complex measuring methods, the collection of this type of data by citizens has not been explored on a scientific basis. However, it is common to include direct measurements of velocity in protocols to monitor the environment and water quality, as it is the case of Hoosier Riverwatch (IDEM, 2015). In these cases, the citizens perform measurements that involve more processing (e.g. definition of transects to measure flow, use of formulas).

To the best of the authors' knowledge, only three studies were found that make use of velocity data collected by citizens, all for the study of floods, as presented in Table 2. Le Boursicaud et al. (2016) evaluated the surface velocity field in a channel from a YouTube video, using the LSPIV methodology (Large Scale Particle Image Velocimetry), an established method to obtain velocity from a sequence of images. For enabling this analysis, information about the camera (model and lens type) is needed, visible, fixed elements are needed to be used as reference points and it is also required that both river banks are visible. Although the method calculates the velocity in two dimensions, in Table 2 we referred to it as 1D because it was carried out in a channel, which in a context of flood modelling is considered as a 1D domain. A complementary project was discussed by Le Coz et al. (2016), in which the same technique is applied to a video crowdsourced by a citizen, this time using the result to estimate discharge and the latter to calibrate a 1D hydraulic model. For this, a visit to the location was needed to extract cross-sectional data. In this context, Yang and Kang (2017) developed a method for crowd-based velocimetry of surface flows, based on Particle Image Velocimetry, in which citizens mark features in the picture. The method has not been tested with citizen collected data yet.

The third study, conducted by Smith et al. (2015), selected Twitter messages that include terms of semantic value related to the citizen location, water depth (e.g. knee-deep) and velocity. The terms were then associated with quantitative values/ranges. The authors did not go into detail on discussing the reliability and uncertainty in such data, even though the issue is recognised.

**Table 2: Scientific literature on citizen contributions to measurement and analysis of velocity**

| Study | Measurement/analysis methods | Type | Purpose | Flood type | Location |
| --- | --- | --- | --- | --- | --- |
| Le Boursicaud et al. (2016) | LSPIV analysis of video collected from social media (YouTube) | 1D | Monitoring | Flash flood | France |

| | | | | |
|---|---|---|---|---|
| Le Coz et al. (2016) | LSPIV analysis of video sent through webpage | 2D Modelling | Fluvial flood | Argentina |
| Smith et al. (2015) | Analysis of texts and pictures collected from social media (Twitter) | 2D Modelling | Pluvial and drainage flood | UK |

## 2.3 Flood extent

Flood extent, similarly to water level, is a variable that is simple to measure as it consists of binary values: flooded or non-flooded area. As a 2D variable, it needs a lot of spatial information and it is the main reason related studies gather flood
5  extent estimates in data rich environments, through social media/photo sharing services mining, as shown in Table 3. In some cases, the citizens act only as sensors, providing pictures to be analysed by the research team, while in other cases they also act as interpreters by providing the flooded/non-flooded information. As can be expected, all studies found were carried out in urban areas.

10  In some of the studies the text and images are indicating the location of their origin as being flooded (georeferenced or inferred) (Aulov et al., 2014; Smith et al., 2015; Yu et al., 2016), whilst in others (Cervone et al., 2016; Li et al., 2017; Rosser et al., 2017; Schnebele et al., 2014; Schnebele and Cervone, 2013) there is processing of the information to infer the surrounding inundated areas. Additionally, the last group of studies mentioned fused flood extent data from citizens with satellite data or with gauge data.

**Table 3: Scientific literature on citizen contributions to measurement and analysis of flood extent**

| Study | Measurement/analysis methods | Purpose | Flood type | Location |
|---|---|---|---|---|
| Cervone et al. (2016), Schnebele et al. (2014), Schnebele and Cervone (2013) | Analysis of pictures and videos collected from social media (Facebook and YouTube) and crowdsourced (Flickr) | Mapping | Flood map | USA and Canada |
| Li et al. (2017) | Analysis of texts and pictures collected from social media (Twitter) | Mapping | Flood map | USA |
| Rosser et al. (2017) | Analysis of crowdsourced pictures (Flickr) | Mapping* | Flood map | UK |
| Aulov et al. (2014) | Visual analysis of texts and pictures collected from social media (Twitter and Instagram) | Modelling | Coastal flood | USA |
| Smith et al. (2015) | Analysis of texts and pictures collected from | Modelling | Pluvial and | UK |

| | | | | |
|---|---|---|---|---|
| | social media (Twitter) | | drainage flood | |
| Yu et al. (2016) | Citizen's visual identification of flooded location collected by governmental Chinese website | Modelling | Pluvial and drainage flood | China |
| Padawangi et al. (2016) | Citizen information | Monitoring | Flood | Indonesia |

\* A statistical model is created, but in this study we consider only physical models in the modelling category

## 2.4 Land cover/Land use

Land cover is not a variable in flood-related models but we include it in this review for its importance in inferring roughness (i.e. the parameter representing momentum loss due to friction, to the ground resistance encountered by the flow). Other

valuable aspects of land use data are the information on roads and structures that can be obstacles to floods, which can be incorporated in the model structure; and the information on vulnerability (e.g. hospitals, dense residential areas, industrial zones), which can be used to obtain flood risk maps. According to Klonner et al. (2016), when reviewing the literature on VGI for natural hazard analysis, there are few studies for vulnerability analysis. The aspects of land use related to vulnerability and risk are complex and study topics on themselves, so these aspects are not discussed further in this article.

Table 4 presents the articles considered for this review. Compared to previously discussed variables, the contribution of citizens to land cover maps generation has been already proved as a concept (Albrecht et al., 2014; Fritz et al., 2012), nowadays being researched further for quality of data (Salk et al., 2016) and fusion of maps (Lesiv et al., 2016).

One of the first publications on the subject was from Iwao et al. (2006), in which they describe the Degree Confluence

Project. The objective was to generate a global land cover map, which implies obtaining ground truth data from around the globe. For obvious reasons, it was unfeasible to make field campaign or analyse low-resolution images with sufficient resolution. Thus, they launched a webpage that invited citizens to visit integer coordinates (e.g. 25° W, 25°) locations, take photos from the four cardinal directions and provide comments on the region. They discovered that citizen-generated data was having quality similar to that provided by specialists.

Another significant project in the area is GeoWiki. It started in 2009 as a platform for people to validate global land cover maps, by comparing their classification to high-resolution images (Fritz et al., 2009). The project has grown since and has recently achieved its main goal: to generate a hybrid global land cover map by fusing existing maps and performing calibration and validation using the analyses made by citizens (See et al., 2015). Current initiatives in the GeoWiki project

include gamification and analysis of pictures uploaded onto the platform (See et al., 2015). Many studies stemmed from the data collected, generally focused on specific land cover types. A similar approach is taken by Dong et al. (2012), that analyses pictures uploaded by citizens using a different web application. The research conducted by Dorn et al. (2014) goes

one step further, as it attributes roughness values to multiple land cover maps, including Open Street Maps ( a website where citizens can modify the current street and land cover map).

**Table 4: Scientific literature on citizen contributions to measurement and analysis of land cover/land use**

| Study | Measurement/analysis methods | Purpose | Flood type | Location |
|---|---|---|---|---|
| Iwao et al. (2006) | Visual interpretation of crowdsourced tagged pictures sent through app/webpage (Degree Confluence Project website) | Mapping | No flooding | Global land cover map |
| See et al. (2015b)* | Visual interpretation of Google Earth and pictures sent through app/webpage (GeoWiki) | Mapping | No flooding | Global land cover map |
| Dong et al. (2012) | Analysis of tagged pictures from Global Geo-Referenced Field Photo Library (DCP citizen pictures + field trip pictures) | Mapping | No flooding | Forest cover map in Asia |
| Dorn et al. (2014) | Use of Open Street Maps | Modelling | Fluvial flood | Austria |

* Many other articles related to crowdsourcing through GeoWiki

### 2.5 Topography

The Digital Elevation Model (DEM) is one of the most important components in flood modelling, as it generally heavily influences flood propagation. It is particularly important in urban settings, where spatial variability in refined scales has a considerable effect on the direction of water flows. Unfortunately, this is a complex variable to measure that so far relies either on fully trained professionals to go to the field, or on expensive airborne technologies. The usage of drones, also called Unmanned Aerial Vehicles (UAVs), is a potential low-cost alternative that is increasingly being more studied (Hamshaw et al., 2017), but so far studies on citizen generated drone data are limited to evaluating the spatial distribution of contributions (Hochmair and Zielstra, 2015) or to the analysis of repositories for image sharing (Johnson et al., 2017). However, recently, Shaad et al. (2016) studied a terrain capturing low-cost alternative to LiDAR remote sensing images and other expensive methods. The low-cost technique is the ground-based close-range photogrammetry (CRP) that consists of collecting images/videos from the ground, post-processing them and obtaining terrain information. Volunteers made the videos in a designated location, where even UAVs would not be able to collect data. After comparing the results to other methods, they concluded that the result has an acceptable quality.

### 2.6 Summary analysis

By classifying the discussed studies according to Craglia et al. (2012), there is an overall similarity in the number of studies that crowdsource data implicitly and explicitly (Fig. 3). It is visible though that this aspect does not translate into

homogeneous distribution per flood-related variables, with most implicitly volunteered contributions being related to flood extent and most explicit being related to water level. There is a slightly higher concentration of modelling studies that are explicitly volunteered, but not enough to be able to draw any conclusions.

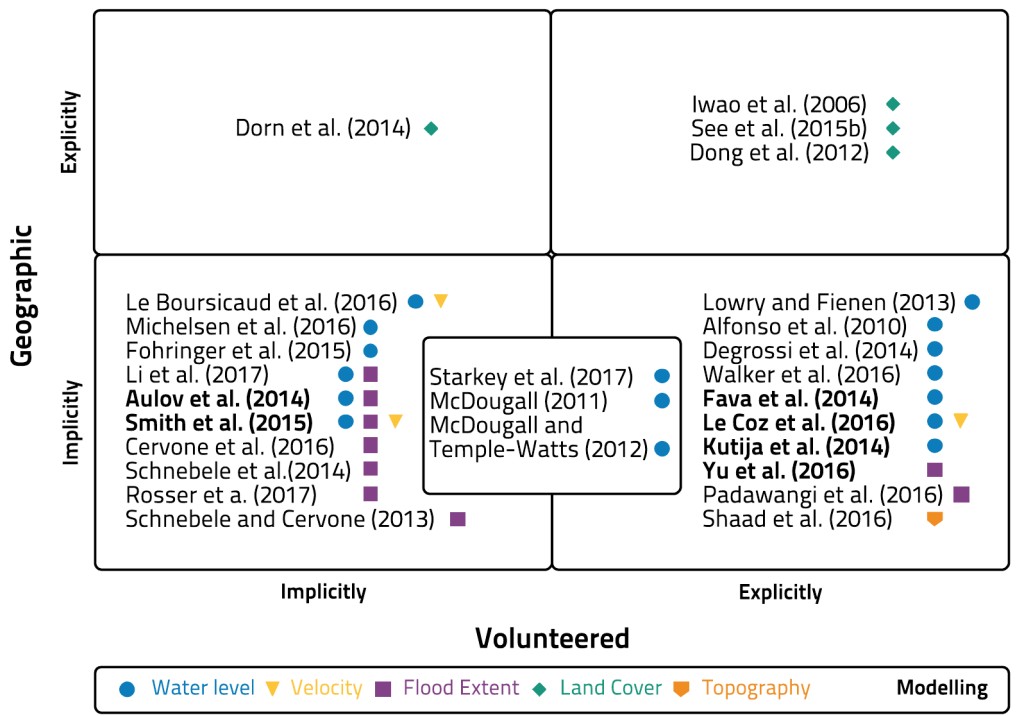

**Figure 3: Selected studies represented in the typology of VGI (Volunteered Geographic Information)**

Considering the temporal distribution of studies evaluated in this review, it is evident that there is a trend: the rise in number of studies from 2014 onwards (Fig. 4). This relates to the initial barrier in acknowledging citizen data as having quality that is high enough for scientific studies (Buytaert et al., 2014). This resistance is reducing over time as such data is being proved useful, protocols are being designed and the data uncertainty is being better understood and quantified.

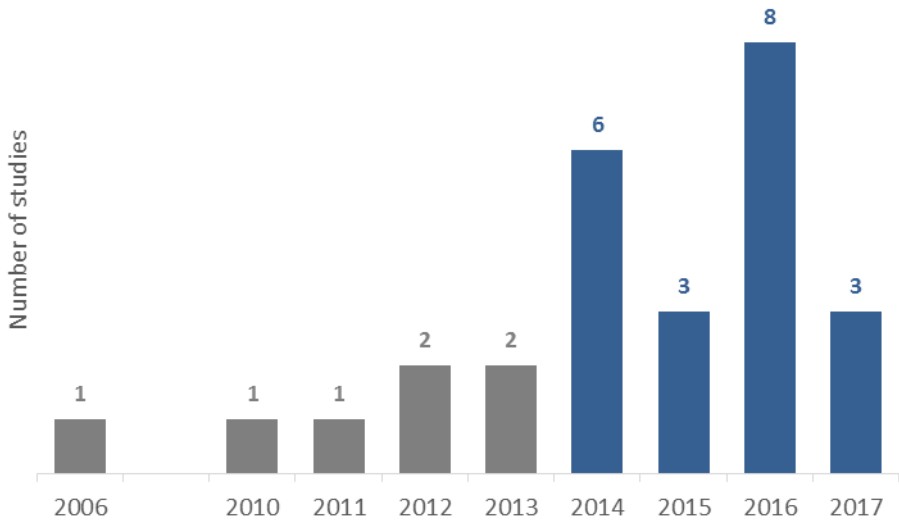

**Figure 4: Number of studies analysed per year**

If the analysed studies are aggregated into categories (Fig. 5), it can be seen that modelling studies amount to approximately

5   the same quantity as monitoring ones, but they are only about a third of all studies reviewed. This is expected because to use data in models it is necessary to monitor them first. Also, monitoring and mapping applications attend to more general end uses. Specifically for land cover, there is an unexplored field in modelling (there are more mapping studies than the ones in the graph, see Sect. 2.4). The reason behind may be that modellers do not tend to validate their own land cover maps and thus will not do it with citizen science data. What can be noted though, is the lack of exploration of velocity and topography

10  variables, which, as mentioned, can be due to the complexity in analysing and setting up the experiment.

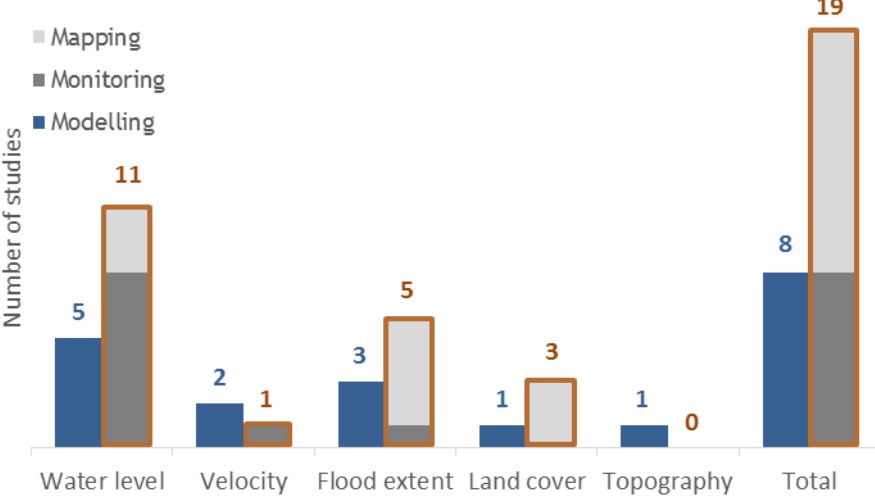

**Figure 5: Number of studies analysed per flood-related variable per category: mapping, monitoring and modelling**

Related to that, previous sub-sections discussed in detail the methods for collection and analysis of flood-related data obtained through crowdsourcing. For example, water level data obtained from reading a water level gauge is easy to collect and easy to analyse. On the other hand, it requires the installation of gauges (Fig. 6). In summary, whenever data is collected from the Internet, there is the disadvantage of needing social media/photo sharing services mining, entailing computational efforts and dealing with a high percentage of data that is not georeferenced or time stamped. Further, in the case of water level and velocity, some studies suggest that also field visits are necessary and the methods to analyse data are complex. Considering crowdsourced data on land cover and topography, it is straightforward to measure and analyse them, although their delivery to the interested parties may require a smartphone app or a website to be set up and maintained (with the exception of Open Street Maps).

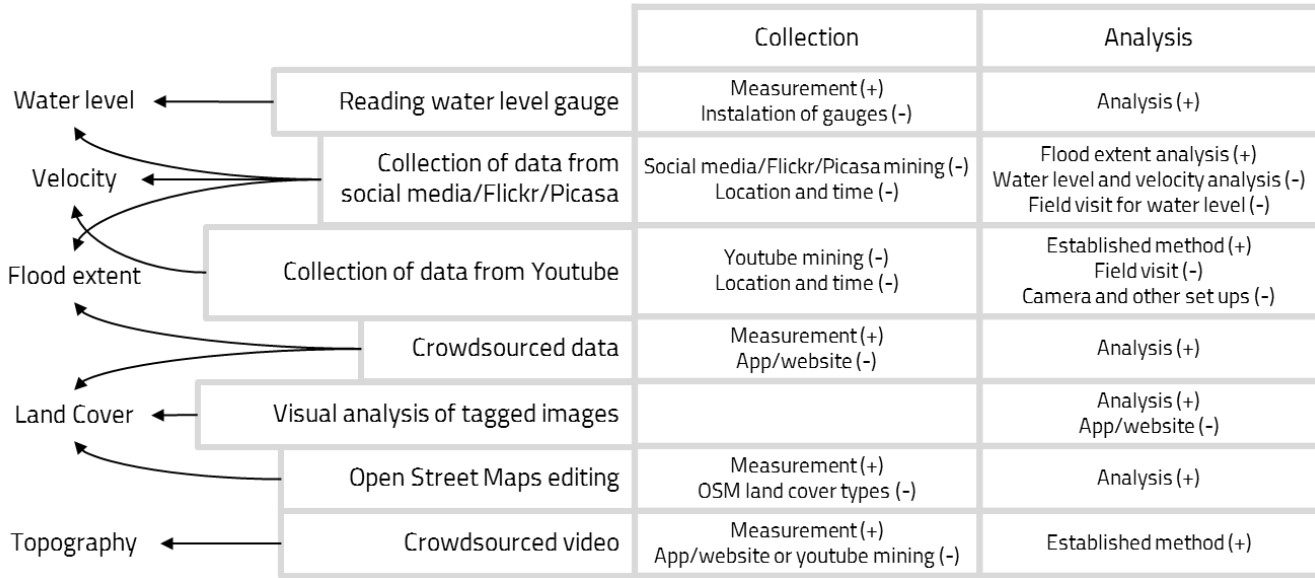

**Figure 6: Pros and cons of collection and analysis methods used to collect flood-related data by citizens**

## 3 Crowdsourced data in flood modelling

By concentrating on the studies in which modelling was performed, we explore in detail how crowdsourced data was
integrated into each component of flood models.

There is a variety of flood models developed to deal with different types of flood, including: fluvial, pluvial, coastal and
drainage floods. The main driver of fluvial floods is upstream river discharge, of pluvial floods is precipitation and of coastal
floods is storm surges. In urban drainage floods, the flows inside, through and outside of drainage systems are pivotal for
flood representation. Moreover, there are complex cases where more than one flood process needs to be represented.
Although in physically-based flood models water flow is computed by the same principles, different sets of data are needed
for different types of flood models. We focus on a general hydrodynamic model definition and its common inputs but present
what was the flood type evaluated in the scientific literature (Table 5).

The flood modelling process typically has two parts: model building, and model usage. (Fig. 7). Model building starts by
defining the model setup (boundary conditions, parameters, schematization, input data), followed by calibration and
validation of the water level and velocity fields (dependent variables) with observed values. Calibration and validation can
be performed for both simulation and forecasting models. Once the model is ready, simulations can be run by using different
boundary conditions or introducing designed measures for better flood management; or forecasts can be made by using
forecasted water levels or discharges as boundaries. In a simulation setting, model parameters are assumed to be constant in
time, while in a forecasting setting the parameters, inputs or states (water levels) can be updated while the model is in use,
using data assimilation.

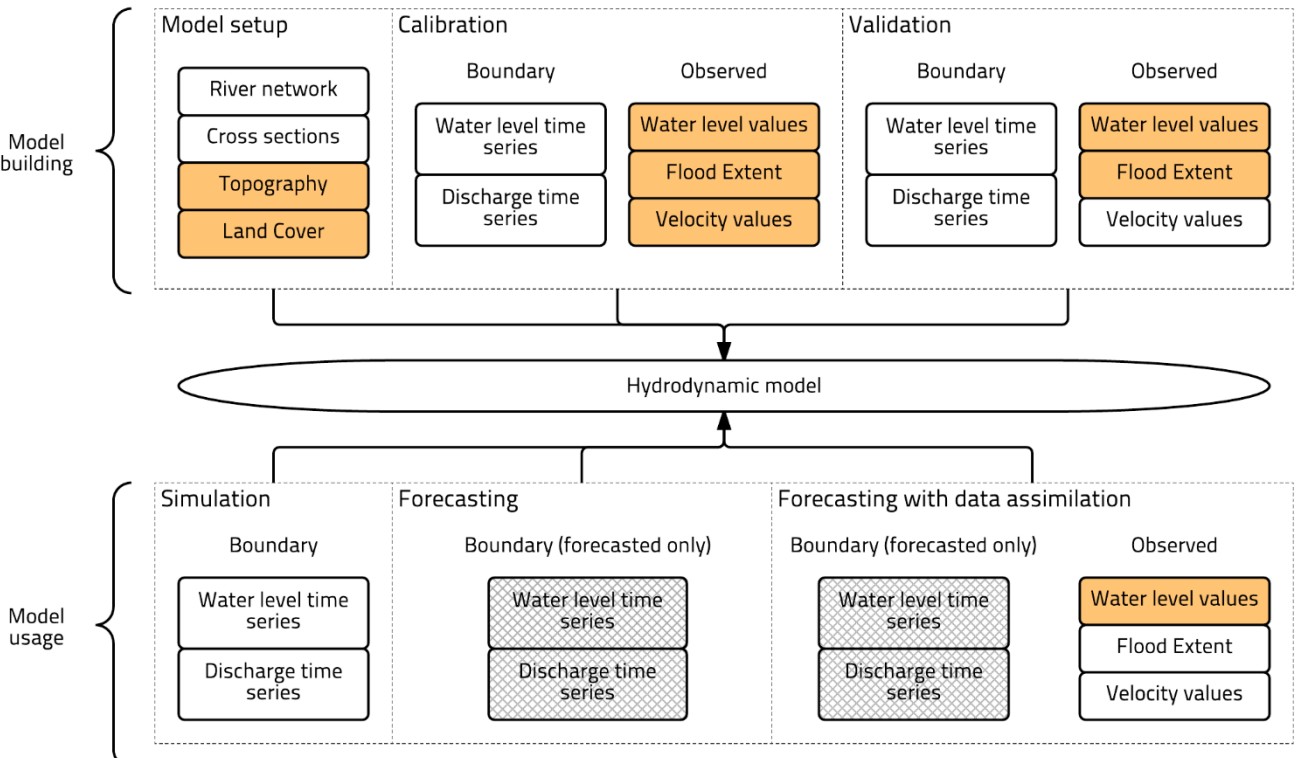

**Figure 7: Flood models data requirements. Orange coloured tiles correspond to data that citizens have contributed to in a flood modelling context and gridded tiles correspond to data citizens cannot contribute to (forecasted water levels and discharges).**

. From the studies analysed (Table 5), three consider 1D channels and the others worked in a 2D setting. Most of them analyse only one variable, except Smith et al. (2015) that evaluate water level and velocity. Moreover, most of them model urban floods, some in a pluvial and others in a fluvial context.

**Table 5: Scientific literature on crowdsourced data used in flood modelling**

| Use in modelling | Study | Measurement method | Type | Variable | Flood type | Location |
|---|---|---|---|---|---|---|
| Model setup | Dorn et al. (2014) | Use of Open Street Maps | 2D | Land cover | Fluvial flood | Austria |
| | Shaad et al. (2016) | Analysis of pictures captured by volunteers at selected location | 2D | Topography | Fluvial flood | Indonesia |
| Calibration | Smith et al. (2015)* | Analysis of pictures and tweets collected from social media (Twitter) | 2D | Water level and velocity | Pluvial and drainage flood | UK |

| | Le Coz et al. (2016) | LSPIV analysis of videos sent through webpage | 1D | Velocity | Fluvial flood | Argentina |
|---|---|---|---|---|---|---|
| | Yu et al. (2016) | Citizen's visual identification of flooded location provided through Chinese website | 2D | Flood extent | Pluvial and drainage flood | China |
| Validation | Kutija et al. (2014) | Analysis of pictures collected from the University and City Council | 2D | Water level | Pluvial and drainage flood | UK |
| | Yu et al. (2016) | Citizen's visual identification of flooded location provided through Chinese website | 2D | Flood extent | Pluvial and drainage flood | China |
| Data assimilation | Aulov et al. (2014) | Visual analysis of texts and pictures collected from social media (Twitter and Instagram) | 2D | Water level and flood extent | Coastal flood | USA |
| | Mazzoleni et al. (2015, 2017) | Simulated citizen reading of water level gauge sent through app | 1D | Water level | Flood forecasting without flood model | Italy and USA |
| | Fava et al. (2014) | Citizen's reading of a water level gauge sent through app or webpage | 1D | Water level | Flood forecasting without flood model | Brazil |

\* It is classified as calibration because, in the classical sense, it improves the model according to observations. However, what actually is done is the fine-tuning selection of the precipitation field that fits the observations better.

Considering model building, specifically the model setup, citizens contributed to improving/updating land cover (and consequently roughness) and topography information. Dorn et al. (2014) used the land cover information contained in Open Street Maps[4] for modelling a fluvial flood. They do not analyse how much contribution was made by the citizens and data processing is restricted to attributing land cover classes to the features displayed in the maps. In the study of Shaad et al. (2016), which addresses topography, there is only one citizen contribution (low-cost alternative) in one selected location that is merged with an existing DEM and then used in the model. In both cases, the objective was to compare the performance of this low-cost alternative against the performance of consolidated technologies when used for hydrodynamic simulations.

Crowdsourced data has also been used to calibrate and validate flood models in four studies. One study gathered such data through social media and public image repositories mining and the others through data uploaded by citizens on specific platforms. Smith et al. (2015) identified storm events through social media, triggering shock-capturing hydrodynamic model

---

[4] Open Street Maps (OSM) is an online platform that provides street maps and other information. The maps provided can be edited by the users at any time

runs with various rainfall intensities. The results were compared with social media data on water level/velocity. The comparison consisted of defining a buffer zone around the crowdsourced observation location, built a histogram of simulated cell values within it and evaluating the overlap of crowdsourced value/range and the histogram 70-95th percentile range. As most citizen contributions did not have a water level/velocity value, they received a minimum water level value. Because of that, the selected simulation was the one with more 'overlaps' and that would not perform better than a simulation with rainfall slightly higher. Yu et al. (2016) collected flooded data through a Chinese website and divided it into calibration and validation data sets for a pluvial flood model verification. There is no mentioning on how this data is provided (e.g. text or image). Le Coz et al. (2016) obtained a discharge value for calibration of a hydraulic model based on the surface velocity data obtained by a video uploaded to a specific website. Kutija et al. (2014) collected pictures uploaded by citizens and extract from them water levels by comparison with reference objects, such as cars (no further detailing on the method of extraction is made). Water level data is then used to validate a pluvial flood model.

The described approaches so far consider citizen data for model building and its possible extension for recalibration and revalidation. Four studies went one step further, integrating crowdsourced data in model usage. Mazzoleni et al. (2015, 2017) used synthetically generated data to represent citizen observations, which were incorporated in the model through data assimilation algorithms, adapted to deal with the intermittent nature of crowdsourced data. Aulov et al. (2014) and Fava et al. (2014) also used the data for simulation/data assimilation, but the methods used are not detailed in the studies. However, the studies of Mazzoleni et al. (2015, 2017) and Fava et al. (2014) were made for flood forecasting through hydrological models and not using hydrodynamic models[3].

### 3.1 Crowdsourced data information content

If we aim at integrating data into model, data accuracy, volume and temporal and spatial coverage should be at a certain level. When these data properties are inadequate, data integration would not provide useful results (i.e. the model performance can be low). Although most modelling variables vary in time and space, the data does not need to cover all dimensions in all parts of the modelling process. For instance, in model setup, topographic data is not needed every 15 minutes, hourly or daily; it can be provided in a discrete time coverage, from months to years. We analyse four data properties: temporal coverage, spatial coverage, volume and uncertainty (Table 6). Although same for all parts, the last two properties vary significantly when analysing the information content of crowdsourced data and that is why these properties are included (Table 6).

**Table 6: Data properties currently required in the modelling process**

| Setup | Calibration & Validation[1] | Simulation | Data assimilation | Data assimilation |
|---|---|---|---|---|
| Topography | Water Level | Water Level | Water Level | Flood Extent |
| Land Cover | Velocity | Velocity | Velocity | |

| Flood Extent | | | | | |
|---|---|---|---|---|---|
| **Temporal coverage** | Discrete | Discrete/Continuous | Continuous | Variable | Variable |
| **Spatial coverage** | Distributed | Discrete/Distributed | Discrete | Discrete | Unknown |
| **Uncertainty** | The lower the better | | | | |
| **Volume** | The higher the better | | | | |

[1] Dependent on purpose of the model

Analysing crowdsourcing studies by their information content, it is possible to draw the following conclusions:

- Model setup: for integration of topographic and land cover data, it is necessary to have spatially distributed data. While this has been achieved within land cover studies, there is only one study involving topography and the data obtained so far have discrete spatial coverage.

- Calibration and validation: through mining and crowdsourcing of water level and flood extent estimates, spatially distributed crowdsourced data have already been obtained for calibration/validation of simulation models. The accuracy of the time stamp was considered vital (Kutija et al., 2015) and results in time have a preliminary good level of agreement with citizen observations (Yu et al., 2016). However, even though these studies compare the results with citizen observations in time, this is done qualitatively and there is no focus on reporting and evaluating the temporal coverage.

- Simulation: traditional modelling efforts require time series of data at specific frequencies, which has only been achieved through crowdsourcing in the realm of community-based approaches, in which water levels are measured at 6 a.m. and 6 p.m. in agreement with the community (Walker et al., 2016). However, this type of data has been only monitored and not used in a modelling context so far.

- Data assimilation: it generally assimilates data provided with a fixed time frequency, but there are a few studies that consider intermittent data to be assimilated (Mazzoleni et al., 2015, 2017). However, similarly to simulation, the temporal coverage of crowdsourced data is insufficient for data assimilation efforts.

Considering uncertainty, this is highly dependent on the collection/analysis method. For example, obtaining water level values from pictures of flooded areas (2D) is uncertain, as it mostly involves the selection of what constitutes a good reference point to be made by the citizen. Flood extent, on the other hand, tends to be less uncertain to measure, due to its binary nature. The collection through data mining (and sometimes crowdsourcing) has, in general, more sources of uncertainty: from geotagging, timestamping and the observed value. To deal with the first two, Aulov et al. (2014) used only data that contained proper geotag and time stamp. Kutija et al. (2014) classified non-timestamped data as during or after the event, based on picture visual inspection, defining an observation time range. Smith et al. (2015) dealt with uncertainty in location by generating a histogram of simulated values around the observed point. Yu et al. (2016) acknowledged these

sources of uncertainty. Regarding uncertainty in value, existent in all sources of crowdsourced data, most studies used the (processed) observations as were, without indication of uncertainty. Smith et al. (2015) defined ranges, although these are not discussed. Mazzoleni et al. (2015, 2017), used uncertain synthetic crowdsourced data with variable uncertainty.

Regarding volume of data collected, this is an issue for all modelling processes, although data mining has again been able to provide a better coverage. Besides the challenge of uncertainty, data mining has also the challenge in providing data in conditions that are not extreme, as most of the contributions are done in floods situations and it is limited to certain variables (water level, flood extent and velocity). Some of the studies were proof of concepts and integrated up to 3 crowdsourced observations each (Le Coz et al., 2016; Fava et al.; 2014; Shaad et al., 2016). Others ranged from 12 to 298 observations

(Kutija et al., 2014; Smith et al., 2015; Yu et al., 2016) and in some cases it was not possible to define the exact number of observations (Aulov et al., 2014; Dorn et al., 2014).

## 4 Opportunities and challenges

In the last years, the interest in citizen science and the number of citizen science studies in the water resources context has risen considerably. The main factors affecting its use in flood modelling are the degree of how difficult it is to acquire and

evaluate these data and their integration into the models. Our analysis of the existing literature allows for pointing out a number of positive experiences from which we can derive opportunities to:

- Explore and improve the existing methods to obtain water velocity and topography from videos
- Explore calibration and validation employing data collected through social media in urban environments
- Explore the possibilities of setting up the models with the use of land cover maps validated with citizen science
- Make use of apps/websites already developed for citizen science

The first one is based on small scale but successful studies related to using well-developed techniques in a citizen science scenario. The relevant experience in data gathering and analysis can be updated to fit the needs of flood modelling. Also,

social media and public image repositories mining has proved to be successful in calibration and validation in modelling studies, proving the concept and opening the opportunity to investigate how large this contribution is. As mentioned previously, in the field of land cover map generation citizen data has been used to validate maps and this successful example could be used to obtain new roughness maps in a modelling context. Lastly, technological development of apps, websites and techniques could be shared and put to public use, to be tested further and to avoid duplicated work.

There are aspects of the integration of crowdsourced data into flood modelling that are still challenging. These are:

- ▪ Explore the use of citizens as data interpreters
- ▪ Improve methods to estimate water level from pictures
- ▪ Harmonise the time frequency and spatial distribution of models with the ones of crowdsourced data
- ▪ Quantification of uncertainty
- ▪ Increase the volume of data gathered, mainly in non-urban environments

Most of the analysed studies regard the citizen as a sensor, with the exception of studies about land cover related data, in which the citizen also acts as an interpreter. For other variables, some studies have already started evaluating the ability of citizens to provide interpreted information (Degrossi et al., 2014), but these are few. Regarding water levels, readings from rulers and extraction from pictures are described differently in the literature, with varying degrees of thoroughness, indicating a need for development and testing of water level measurement methodologies in the context of citizens' contributions. The third point brings up a challenge that concerns not only citizen science but also modelling: what is the necessary temporal and spatial distribution? Is the traditional modelling approach definitive in terms of data requirements and citizen science approaches should adapt to it, or, the modelling process can be adapted to receive citizen science data? The fourth challenge relates to the quality of data and, again, in the area of global land cover maps some articles have already discussed the subject (Foody et al., 2013), but still, when modelling is concerned, the crowdsourced data are treated as traditional data and the issue of quality is hardly addressed (albeit recognized as an issue). To which extent does this assumption hold? What is the uncertainty in citizen science data? Lastly, there is a challenge mentioned by many studies but not really addressed in itself and it is the volume of data. Although the volume of data necessary depends on the objective of the modelling effort, the volume of crowdsourced data tends to be low, lacking temporal/spatial coverage for integration into models. This leads to the question: How to increase the volume of data? Considering this limitation, it is also natural to move towards the question: How much data is needed to improve the model significantly?

Application of citizen science in modelling brings an extra challenge of interdisciplinary. Among similar technical fields (e.g. geosciences and hydrodynamic modelling) there is an issue of technology transfer to be addressed, and there are discussions on underlying assumptions and uncertainties that need to be considered. Additionally, hard and soft sciences are also very linked, as the quality and value of the citizens' observations and their temporal/spatial coverage are intrinsically related to social drivers such as why citizens engage, for how long, with which frequency and what is the role of various stakeholders.

## 5 Conclusions and recommendations

Citizen science has successfully made its way in many scientific domains and it is only fair that the contribution of citizens to modelling floods is also investigated, due to the related intensive data needs. Analysis of literature clearly shows an

increasing number of scientific studies in this area. Successful examples of using existing measurement and analysis methods (e.g. velocity and land cover) and of modelling floods with citizen science data (e.g. social media mining) have been published and are seen as a good basis for further exploration. There is a clear need to standardise and consolidate methodologies and there are challenges involving temporal and spatial distribution of data, uncertainty and volume.

It can be observed that the role of citizen contributions is not only in providing information about the current state of the environment, in monitoring and mapping studies, but also in providing data that can be used in its modelling and forecasting. Studies reviewed in this article showed that crowdsourced data can be integrated: in model building, to improve their overall performance; and directly into models (by data assimilation), to improve immediate forecasts. These are promising studies, however still too few, and they highlight the need for further work in this direction. The integration of crowdsourced data into flood models is a viable way to help solve issues of data scarcity, with a higher potential in ungauged catchments and systems subject to change (e.g. climate change).

One of the challenges worth mentioning is the integration of citizen data with other more traditional data sources like gauging and remote sensing. It is also necessary to analyse cases in which citizens are involved at higher levels of engagement (e.g. participating in the problem definition, analysis of results and even in the decision-making process) and to evaluate the trade-off between model data needs and levels of engagement. The active involvement of citizens may lead to more data collected, which in turn, may lead to more involvement and subsequently, to improved modelling of floods.

Finally, there is the challenge to make citizen contributions valuable in a time where automation in gaining increasing space. One may say that citizens are not needed because of automated sensors. At the same time, there are situations where crowdsourced data are very valuable. One of the non-technical challenges that we see here is to demonstrate such situations and increase acceptance of crowdsourced data by water managers.

**Acknowledgements**

This work was carried out with the partial funding from the Horizon 2020 European Union project SCENT (Smart Toolbox for Engaging Citizens into a People-Centric Observation Web), under grant number 688930.

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
