# Peer review of "Citizen observations contributing to flood modelling: opportunities and challenges"

_Hydrology and Earth System Sciences, 2017_

## Short Comment (SC1) · 6 Oct 2017

The review of how citizen observations have been used in flood modelling research is useful and very timely. The main value of the review is in mapping out the different case studies, identifying trends, and pointing out research gaps. Minor revisions are recommended:

.

Page 1 line 27:

- Do the authors refer to the general need for data in modelling, or specifically to monitoring data used for calibrating the models?

[Figure]

- The example in the second sentence "This is especially true.." requires some explanation.

.

Page 3 line 26:

- Effort is made to present two classification systems. However, these classifications are not used in sections 2 and 3.

.

Page 4 line 10:

- It is unclear why geo-tagged information is not explicitly geographic.

.

Page 4 lines 15-20:

- It does not seem appropriate that SCENT is given a prominent position in this review paper, which should review published literature and not ongoing projects.

.

Page 5 Figure 2:

- Fig 2 illustrates nicely how specific examples are classified within Craglia et al.'s definition, and therefore more examples would be beneficial. It would be even better if the examples were taken from literature.

- SCENT should be removed from the figure.

- it is unclear why the CAPTCHAs are neither implicit nor explicit

.

Page 6, lines 1-2:

- Have studies such as Merkuryeva et al. (2015) been included in the review? please specify.

- The citation is not necessary

.

Page 6, line 18-20:

- It is unclear why the text example is provided in the same paragraph as the images/videos and not in the previous paragraph.

.

Page 7, Table 1:

- It would be good to split the column 'case study' into two columns 'location' and 'flooding type'

- What ordering is used in the table? publication year might make sense.

.

Page 12, Figure 3:

- The review extends to April 2017 - has the publication count for the year 2017 been normalized?

.

Page 13, line 7:

- Flickr and Picasa are products, it is better to refer to photo sharing services.

- what is exactly meant with 'mining', and how does that entail low-quality data?

.

Page 18, lines 19-25:

- The discussion on reliability and volume of data is interesting and necessary, but the statements do not seem to make good of the review that was conducted. Do none of the papers attempt to quantify uncertainty?

- Is the volume of data per type (water level, velocity, etc.) available comparable to the number of case studies?

.

Page 19, lines 20-26:

- The language used is imprecise.

- "interactions between citizen science and water resources"

- "Deal with uncertainty"

---

## Author Comment (AC1) · 18 Oct 2017

**SC1**: The review of how citizen observations have been used in flood modelling research is useful and very timely. The main value of the review is in mapping out the different case studies, identifying trends, and pointing out research gaps. Minor revisions are recommended:

**AC**: Thank you for finding the paper timely and for the appreciation of the review paper. Authors would like to thank M. Moy de Vitry for taking time to review the paper and add to the ongoing discussion. The comments and suggestions received are of high value, and based on them we will make improvements to the manuscript. Please see below the answers to the comments.

[Figure]

**Page 1 line 27:**

**SC1**: Do the authors refer to the general need for data in modelling, or specifically to monitoring data used for calibrating the models?

**AC**: Authors are referring to general data needs for modelling floods, no special distinction for calibration is made. Thank you for pointing out the confusion. In order to clarify this issue to the reader an additional statement will be added to the manuscript (the added text is highlighted in bold):

"In order to have adequate representation of floods, most models require large amounts of data, **both for model building and model usage.**"

**SC1**: The example in the second sentence "This is especially true.." requires some explanation.

**AC**: More explanation will be added, by rephrasing the sentence (in bold):

"This is especially true for pluvial flood modelling **, where flooding may not occur in gauged rivers and hence, flow gauging stations outside of flooded zones may be of little use.**"

**Page 3 line 26:**

**SC1**: Effort is made to present two classification systems. However, these classifications are not used in sections 2 and 3.

**AC**: These classifications are not introduced for the purpose of further classifying other papers, but for opening the discussion and debate on the existing reviewed literature. The first classification system (i.e. level of engagement), aims to explicitly say that discussion on advantages/disadvantages of collection/analysis methods, as well as their purposes, is strictly addressing contributions in terms of quantitative data (i.e. contributions towards flood modelling); and it does not address the advantages/disadvantages

of contributions from other types of involvement. For example, it is out of the scope of the article to discuss tacit knowledge or social media mining having the (possible) disadvantage of not fostering awareness. For further clarification, the next version of the manuscript will be amended with the following text (the one highlighted in bold):

"The aim of the review presented in this current article is focused on the contribution towards flood modelling only, coming most prominently from the two lowest levels of engagement. **We do not discuss topics related to engagement in the generation of (quantitative) data.**"

The second classification system was made to provide a reflection of such components (implicitly/explicitly geographic and implicitly/explicitly volunteered) when data is obtained from citizens. Based on this and a follow-up comment, we will add a Figure where we place on the framework the studies cited in this paper; and we will provide an analysis of such result.

**Page 4 line 10:**

**SC1**: It is unclear why geo-tagged information is not explicitly geographic.

**AC**: For clarity, in the beginning of such paragraph the following phrase will be added (in bold):

"Another way to classify citizen science initiatives (within the context of VGI) is by setting them as implicitly/explicitly volunteered and implicitly/explicitly geographic (Craglia et al., 2012). **In this classification system, geographic refers to the main information conveyed through the contributed data, therefore, geo-tagged data is not necessarily geographic.**"

**Page 4 lines 15-20:**

**SC1**: It does not seem appropriate that SCENT is given a prominent position in this

review paper, which should review published literature and not ongoing projects.

**AC**: As mentioned in the acknowledgements, this review and research related to it are supported by the H2020 project, SCENT. Therefore, it is natural that the ideas generated within the project, which aims at covering scientific gaps, are properly acknowledged in the paper text as well. The inclusion of SCENT has as objective to illustrate the classification system, taking advantage of the fact that in the project the four classes are being covered. For clarity, it was not chosen to include published literature in this part of the article without analysing it first. As per suggestion of the reviewer, we will present the same scheme later on, where such literature will be included.

**Page 5 Figure 2:**

**SC1**: Fig 2 illustrates nicely how specific examples are classified within Craglia et al.'s definition, and therefore more examples would be beneficial. It would be even better if the examples were taken from literature.

**AC**: Thank you for this suggestion, we will take it into account and expand in the second version of the manuscript.

**SC1**: SCENT should be removed from the figure.

**AC**: The justification of SCENT's inclusion in the figure has been provided in a previous comment. This figure sets the scene for the second one that will be added based on the reviewer's suggestion.

**SC1**: it is unclear why the CAPTCHAs are neither implicit nor explicit.

**AC**: In the image CAPTCHA plugin is both implicit and explicit. The text will be modified for clarification (in bold):

"It lies in the middle of this quadrant as it encourages citizens to participate in gaming to

collect land cover/use data, in field campaigns to collect other implicitly geographic information (e.g. water level), and also aims to obtain implicitly volunteered contributions through a CAPTCHA plugin, in which citizens tag images **, e.g. of land cover/use or water level,** in order to access online content".

**Page 6, lines 1-2:**

**SC1**: Have studies such as Merkuryeva et al. (2015) been included in the review? please specify.

**AC**: No, they have not been included. The text will be modified for clarification (in bold):

"It needs to be noted that there are studies **that were not included in the present review because** they just mention the use of crowdsourced data and do not provide more relevant information on collection, analysis and quantity of data, such as Merkuryeva et al. (2015). **The same is the case of** studies that evaluate variables qualitatively, in ways that cannot be directly associated with modelling (Kim et al., 2011)."

**SC1**: The citation is not necessary.

**AC**: We acknowledge that the citations do not serve a purpose other than being examples. However, as a review paper, we consider that different aspects of the literature should at least be exemplified, in a way that the interested reader may wish to explore topics not covered in the review.

**Page 6, line 18-20:**

**SC1**: It is unclear why the text example is provided in the same paragraph as the images/videos and not in the previous paragraph.

**AC**: Thank you for bringing up this misunderstanding. The text examples are related to non-quantitative text that is converted to quantitative measures. As the section's first

paragraph is about quantitative crowdsourced data and the second is about qualitative ones, this information fits better in the second paragraph. For clarity, the paragraph will be rephrased as follows (in bold):

"In other cases **, the citizen provides qualitative data that will be compared to references by researchers**. Mostly during flooding situations, citizens provide pictures (Fohringer et al., 2015; Kutija et al., 2014; Li et 15 al., 2017; McDougall, 2011; McDougall and Temple-Watts, 2012; Smith et al., 2015; Starkey et al., 2017) or videos (Le Boursicaud et al., 2016; Le Coz et al., 2016; Michelsen et al., 2016). In the case of pictures/images, the water level is compared with objects in the images that have known or approximately known dimensions. For videos, although water level was estimated, the main goal was to obtain discharge values, via estimates of flow velocity. In two cases, texts from citizens were used **(e.g. water over the knee)**, to **calculate** water level values or assuming a certain value when no value was provided (Li et 20 al., 2017; Smith et al., 2015). This sort of data (text, pictures and videos) was mostly collected through social media and public image repositories, requiring mining of the relevant material and dealing with uncertainties in the spatio-temporal characterization of the data of interest."

**Page 7, Table 1:**

**SC1**: It would be good to split the column 'case study' into two columns 'location' and 'flooding type'

**AC**: The columns will be split and studies with flood-related crowdsourced but without floods will be classified as 'No flooding'.

**SC1**: What ordering is used in the table? publication year might make sense.

**AC**: The ordering used in the table was done by grouping papers with similar measurement/analysis methods, followed by the order monitoring, mapping and modelling.

This is similar to the way the analysis is done.

**Page 12, Figure 3:**

**SC1**: The review extends to April 2017 - has the publication count for the year 2017 been normalized?

**AC**: No, it has not been normalized. We understand your reasoning, but our focus is on the content and interpretation, not on a precise, numerical analysis of the contributions. Thus, because of the small numbers of contributions per type of publications, for simplicity, we decide not to normalize.

**Page 13, line 7:**

**SC1**: Flickr and Picasa are products, it is better to refer to photo sharing services.

**AC**: We will change to the proposed terminology.

**SC1**: what is exactly meant with 'mining', and how does that entail low-quality data?

**AC**: Mining refers to the extraction of specific data from a dataset. For example, tweets can be mined from Twitter for a certain period of time and for tweets that contain the word 'flood'. We will expand the first appearance of this term to include such qualification and make it clearer (page 6, lines 20-22, in bold):

"This sort of data (text, pictures and videos) was mostly collected through social media and public image repositories. **Gathering data from such sources requires** mining of the relevant material **(i.e. extraction of specific data from the overall dataset)** and dealing with uncertainties in the spatio-temporal characterization of the data of interest."

Crowdsourced mined information has the possibility of not having a precise time-stamp

or geotag. Thus, there is uncertainty related to it. We consider that the higher the uncertainty, the lesser the quality of the data.

**Page 18, lines 19-25:**

**SC1**: The discussion on reliability and volume of data is interesting and necessary, but the statements do not seem to make good of the review that was conducted. Do none of the papers attempt to quantify uncertainty?

**AC**: Yes, some of the papers do. We will expand this discussion to include more information.

**SC1**: Is the volume of data per type (water level, velocity, etc.) available comparable to the number of case studies?

**AC**: We have not computed the volume of data for each data type. At the moment we estimate that they are directly proportional to the number of case studies. Unfortunately, it is not possible to get the exact number as in some cases more than one variable is collected and no distinction in the overall count is provided.

**Page 19, lines 20-26:**

**SC1**: The language used is imprecise.

**AC**: The language will be rephrased. See below.

**SC1**: "interactions between citizen science and water resources"

**AC**: Will be rephrased as (in bold):

"There are aspects of **the integration of crowdsourced data in flood modelling** that are still challenging."

**SC1**: "Deal with uncertainty"

**AC**: Will be rephrased as (in bold):

"**Quantification of** uncertainty"

---

## Referee Comment (RC1) · Anonymous Referee #1 · 30 Oct 2017

Dear editor, I went through the paper entitled "Citizen observations contributing to flood modelling: opportunities and challenges" by Assumpcao et al. Bringing people's idea and their involvement in science (citizen science) is becoming significant globally. This paper is exactly what lies behind the role of citizen science in combating the flooding by modelling. However, I find the paper is quite difficult to follow in its current form. This also has no such in-depth assessment of the role of citizen science in mitigating climate-induced flood events/hazards. The synthesis/review would have been much useful and interesting if this were focused on one or two key objectives. For example, how citizen science would link to model building process based on crowdsourced data and how citizens themselves would be benefitted provided the feedbacks for the model improvement. Some specific comments Page 2 Line 10-15 what are the valuable con-

tributions? elaborate Page 2 Line 22-26 what are three projects? provide the summary Page 4 Line 19 please define 'CAPTCHA plug in framework', not all readers would necessarily know about it Page 10 Line 12-17 what level of citizens will get involved to generate data globally as many citizens are devoid of IT technology? Page 15 Fig. 6 perhaps Fig. 6 holds the core concept of the paper, where the citizen science link to modelling and its application Page 18 Line 23 please provide what consequences of uncertainty in data mining and how this is improved?
* * *

---

## Referee Comment (RC2) · Anonymous Referee #2 · 30 Oct 2017

This paper present an interesting and fairly complete review on the use of crowdsourcing for flood modelling purposes. The effort to try and characterise the reliability and uncertainty associated to different types of data and different methods of involving citizens in collected them is worth highlighting. I would limit my review to three general comments: (1) There is no mention in the paper of the diversity of models that are used for flood modelling, and whether they are more or less suited for integrating the different types of citizen observations. Arguably, one of the challenges for hydrologists could be to design models specifically for that purpose. At least, it would have been interesting to have some information of the kind of models used in the studies analysed in the paper. (2) the question of time is only very briefly discussed, while in flood modelling, and particularly for real time flood forecasting, this is an critical issue: models not

only require the highest water level or the maximal flooded area extension (which are, I guess, when most of the observations are done), but high resolution data during the rising part of the hydrograph. What have been done to collect this information, and/or what type of participatory approach should be organised to do so? (3) In the same line, rainfall is almost absent in the discussion. As far as I know, crowdsourcing have also been used to obtain spatially distributed rainfall, and many extreme storm events are characterised by a high spatial variability of rainfall, so I suspect that this type of citizen observation could be useful.

---

## Referee Comment (RC3) · Anonymous Referee #3 · 14 Nov 2017

This paper addresses a very timely and interesting topic: citizen science and its use in flood modelling. It will provide some guidance to researchers struggling with the lack of traditional data and at the same time resistant to adhere to alternative data sources. Overall, the text is rather fluid and well written, but in topic 3, "crowded source data in flooding modeling", the explanation of some uses of citizen data in modeling is confusedly described and could benefit from a restructuring of description of uses. Also, despite the relatively large number of papers gathered, the revision process and papers selection is not fully described. Thus, for a synthesis paper, it will be worth proving a perspective on how exhaustive were the efforts undertaken in the collection and selection of relevant studies, and the data sources consulted.

A few minor points include:

[Figure]

In Figure 1, only level one is termed crowdsourcing, not level 2, as stated in the text (page 3, lines 30-31).

It is not clear how the CAPTCHA plug in works as a volunteered contribution; please provide a better explanation.

Figure 2 does not seem relevant, I suggest excluding it; while Figure 6, in its present form, does not seem very informative.

I suggest merging Section 1.2 - Article outline with the end of the Introduction (page 2, line 30).

There are some unnecessary wording throughout the paper, for example: "We have seen in the previous section that" and "In this section we intend to" (page 14, lines 4-5).

---

## Referee Comment (RC4) · Anonymous Referee #3 · 15 Nov 2017

The issue concerning the CAPTCHA plug in is not really about its definition, but about HOW it will be used as a "volunteered contribution". How, for example, random images of a deforested area chosen for security reasons will contribute to the monitoring of land use?

Additionally, despite the content of the outline, it makes more sense that it comes at the end of the introduction, providing readers with an initial and general idea of what will follow. The current topic 1.1 could come together with the description of data sources used and papers selection (to be included), as part of a methodology section.

---

## Author Comment (AC2) · 15 Nov 2017

We thank the reviewer for taking the time to review this paper and for providing useful feedback. Your input is valuable in improving the scientific quality of the paper and its readability. Please find below our answer addressing your comments.

**Comment #1:** Dear editor, I went through the paper entitled "Citizen observations contributing to flood modelling: opportunities and challenges" by Assumpcao et al. Bringing people's idea and their involvement in science (citizen science) is becoming significant globally. This paper is exactly what lies behind the role of citizen science in combating the flooding by modelling. However, I find the paper is quite difficult to follow in its current form. This also has no such in-depth assessment of the role of citizen

[Figure]

science in mitigating climate-induced flood events/hazards.

**Authors' response:** We acknowledge that the assessment of the role of citizen science in mitigating climate-induced flood events/hazards is not addressed in the present article and that is because the focus of the paper is different, particularly it is to review the existing scientific literature regarding the actual and potential crowdsourced data for *flood modelling*. From that perspective, climate-induced flood events/hazards do not bring different challenges for citizens' data collection compared to "regular" flood events. Of course citizen science is much broader than only crowdsourcing of data, but such broad perspective is outside of the scope of this article. Regarding mitigation, in the article's Introduction (page 2), we are mentioning the review of Horita et al. (2013); and the studies of Dashti et al. (2014) and Oxendine et al. (2014). They are addressing disaster management and damage data collection, including the role of citizen science for mitigation of floods in general. In the present article the analysis is made for model improvement, but the model may have multiple purposes (e.g. flood risk or ecosystem conservation). The paper determines what are the benchmarking difficulties and benefits of collecting flood-related data by citizens and of integrating them into models, for the purposes of model set up, calibration, validation, simulation and forecasting.

Given your comment, we realise that an improved explanation of the aim of the review presented in the article is required in the Introduction section. The revised version of the manuscript will emphasize this aspect.

**Comment #2**: The synthesis/review would have been much useful and interesting if this were focused on one or two key objectives. For example, how citizen science would link to model building process based on crowdsourced data and how citizens themselves would be benefitted provided the feedbacks for the model improvement.

**Authors' response:** The approach taken was to group and analyse the studies in which crowdsourced data was integrated into each part of the flood modelling process.

We could not take a different approach because unfortunately the literature on specific parts is scarce (e.g. in Table 5, Page 16, we found just 6 studies on the model building process). Hence, the review could not be limited to few particular aspects. Similarly, although a paper on citizen's benefits from model improvement would be useful and interesting, this is a recent topic that has not been explored enough and there are not enough publications to date so that a review is required or can be made.

**Some specific comments:**

**Page 2 Line 10-15:**

**Comment #3:** what are the valuable contributions? elaborate

**Authors' response:** As suggested we will elaborate in the next version of the manuscript. For example, the CITI-SENSE project managed to simultaneously collect perception data and acoustic measurements in an approach that can be used to develop citizen empowerment initiatives in case of noise management (Aspuru et al., 2016).

**Page 2 Line 22-26:**

**Comment #4:** what are three projects? provide the summary

**Authors' response:** The manuscript will be changed to include such a summary.

**Page 4 Line 19:**

**Comment #5:** please define 'CAPTCHA plug in framework', not all readers would necessarily know about it

**Authors' response:** The following footnote will be added to the manuscript in order to

clarify the concept of a CAPTCHA plugin:

"CAPTCHA stands for 'Completely Automated Public Turing test to tell Computers and Humans Apart'. It is a test evaluating if the subject is human, which is used in websites to provide security. After the test is done the user can be asked to perform extra tasks, for example, tag images."

**Page 10 Line 12-17**

**Comment #6:** what level of citizens will get involved to generate data globally as many citizens are devoid of IT technology?

**Authors' response:** Iwao et al. (2006) did not provide any information on the profile of citizens, nor on engagement strategies, although the lack of data in certain regions was shortly addressed. However, as stated in the Citizen Science section of the manuscript (page 3), the review did not discuss the mechanisms of citizen engagement and participation, as this is a research topic on its own and we focus on data integration. To address this issue, also raised by a comment of a reader in the HESSD interactive discussion, the paper will be modified as follows (modifications are highlighted in bold text):

"The aim of the review presented in this current article is focused on the contribution towards flood modelling only, coming most prominently from the two lowest levels of engagement. **The review does not discuss topics related to engagement for the generation of (quantitative) data**."

**Page 15, Fig. 6:**

**Comment #7:** perhaps Fig. 6 holds the core concept of the paper, where the citizen science link to modelling and its application

**Authors' response:** Though the figure is a core concept of the paper, the paper structure is such that first the wider scope of the paper is defined, laying all the literature that has the potential to contribute to flood modelling in terms of flood-related data. This literature is characterized and analysed for advantages and disadvantages. Then, it is presented an in-depth analysis of the scientific contributions to each part of the modelling cycle. The existing literature is evaluated in terms of its information content and analysed to check how much it matches model requirements. Finally, opportunities and challenges are identified. Following this structure, Figure 6 is presented in a later section.

**Page 18 Line 23:**

**Comment #8:** please provide what consequences of uncertainty in data mining and how this is improved?

**Authors' response:** The consequence of uncertainties, including the ones of data mining, is low model performance. We consider that the higher the uncertainty, the lesser the quality of the data. Hence, because data obtained through data mining has, in general, more sources of uncertainty (from value, geotagging and timestamping), they can potentially be of lesser quality and result in models with low performance. As suggested by a reader, this will be further extended in the next version of the manuscript. To date, in modelling studies, there are only few studies that quantify the uncertainty from crowdsourced data, the impact on model performance or that consider methods for its reduction. To remain neutral, we did not include in the manuscript anything beyond what is in the literature, thus we do not include a discussion on how to improve the situation in modelling.

---

## Author Comment (AC3) · 15 Nov 2017

We would like to thank the reviewer for the revision. We appreciate the comments provided, that deliver insightful and enriching recommendations on how to improve the content of the paper. We have addressed your comments individually in the text below.

**Comment #1:** This paper present an interesting and fairly complete review on the use of crowdsourcing for flood modelling purposes. The effort to try and characterise the reliability and uncertainty associated to different types of data and different methods of involving citizens in collected them is worth highlighting. I would limit my review to three general comments: (1) There is no mention in the paper of the diversity of

[Figure]

models that are used for flood modelling, and whether they are more or less suited for integrating the different types of citizen observations. Arguably, one of the challenges for hydrologists could be to design models specifically for that purpose. At least, it would have been interesting to have some information of the kind of models used in the studies analysed in the paper.

**Authors' response:** The manuscript will be modified to include an explanation on types of flood models (fluvial, pluvial, coastal and drainage). The matter of suitability is not addressed, mainly because the considered papers are not addressing the suitability. However, we found this comment very valuable and we will add more information on the models used in the reviewed studies.

**Comment #2**: (2) the question of time is only very briefly discussed, while in flood modelling, and particularly for real time flood forecasting, this is an critical issue: models not only require the highest water level or the maximal flooded area extension (which are, I guess, when most of the observations are done), but high resolution data during the rising part of the hydrograph. What have been done to collect this information, and/or what type of participatory approach should be organised to do so?

**Authors' response:** In the section 'Crowdsourced data information content' on pages 17-18, we discuss the question of time within each part of the flood modelling cycle. Flood forecasting is not included because citizens cannot provided forecasting data. We acknowledge that we do not consider calibration and validation for specific purposes and thus do not consider them done specifically for obtaining an operational model for flood forecasting. With that in mind and in view of the reviewer's comment, we will change Table 6, in the column 'Calibration Validation', the temporal coverage to 'Discrete/Continuous' and the spatial coverage to 'Discrete/Distributed'. A remark will be added to the table mentioning that the data properties for calibration and validation depend on the purpose of the model. Moreover, the discussion on page 18 and line 8 is extended to accommodate such view and answer the question on what has been done

to collect this time sensitive information. Organisation of participatory approaches are not discussed as they are outside the scope of the proposed article.

**Comment #3:** (3) In the same line, rainfall is almost absent in the discussion. As far as I know, crowdsourcing have also been used to obtain spatially distributed rainfall, and many extreme storm events are characterised by a high spatial variability of rainfall, so I suspect that this type of citizen observation could be useful.

**Authors' response:** We agree that contextualization of the rainfall component is lacking and this will be added to the manuscript. We will mention its importance for certain types of flooding and will provide pointers to articles on crowdsourced data for rainfall. We acknowledge that citizen contributions could be useful for observation of this variable, however, we will not include rainfall in the flood-related crowdsourced data section because it is already covered by the review of Buytaert et al. (2014). Rainfall is a variable of greater importance for hydrological models, whilst the review focusses on a hydrodynamic representation of floods.

---

## Author Comment (AC4) · 15 Nov 2017

We thank the reviewer for providing feedback on the quality of the paper. The review is valuable for making the paper clearer and more structured and the comments are highly appreciated. Please find below our response to the provided comments.

**Comment #1:** This paper addresses a very timely and interesting topic: citizen science and its use in flood modelling. It will provide some guidance to researchers struggling with the lack of traditional data and at the same time resistant to adhere to alternative data sources. Overall, the text is rather fluid and well written, but in topic 3, "crowded source data in flooding modeling", the explanation of some uses of citizen data in modeling is confusedly described and could benefit from a restructuring of description

[Figure]

of uses.

**Authors' response:** By suggestion of the reviewer, the description of uses in Section 3 will be restructured for the next version of the manuscript.

**Comment #2**: Also, despite the relatively large number of papers gathered, the revision process and papers selection is not fully described. Thus, for a synthesis paper, it will be worth proving a perspective on how exhaustive were the efforts undertaken in the collection and selection of relevant studies, and the data sources consulted.

**Authors' response:** The manuscript will be extended to inform that the papers' collection was done through multiple platforms (e.g. ScienceDirect and GoogleScholar), exemplifying used keywords. Additionally, explanation on the selection criterion for consideration will be given, which is the generation/use of flood-related crowdsourced data. The first paragraph of Section 2 (pages 5-6) gives an explanation on how the selection of relevant studies was done (i.e. why certain articles were not selected).

**A few minor points include:**

**Comment #3:** In Figure 1, only level one is termed crowdsourcing, not level 2, as stated in the text (page 3, lines 30-31).

**Authors' response:** The sentence will be rephrased as follows (modifications to the manuscript's text are highlighted in bold):

"Further in this article, **for readability, only the term crowdsourced data is used to refer to data from these two levels of engagement**."

**Comment #4:** It is not clear how the CAPTCHA plug in works as a volunteered contribution; please provide a better explanation.

**Authors' response:** Clarification regarding the CAPTCHA plug-in will be done by means of a footnote, as also requested by another reviewer. The footnote will read as follows:

"CAPTCHA stands for 'Completely Automated Public Turing test to tell Computers and Humans Apart'. It is a test evaluating if the subject is human, which is used in websites to provide security. After the test is done the user can be asked to perform extra tasks, for example, tag images."

**Comment #5:** Figure 2 does not seem relevant, I suggest excluding it; while Figure 6, in its present form, does not seem very informative.

**Authors' response:** Figure 2 was included as an introductory example of framework for analysing crowdsourced data. We acknowledge that it does not attend other purposes in the current version of the manuscript. As per suggestion of a reader that commented on HESSD interactive discussion, we will include a modified version of Figure 2 further in the text, changed to include the reviewed literature. The motivation behind increasing the relevance of such a figure is two-fold: exposition to the interested reader of classification systems of citizen science approaches; connect at a superficial level with social studies that evaluate these classifications, to increase the integration among disciplines.

Figure 6 presents visually two types of information: the components of the flood modelling process and the data necessary for each component; citizen contributions within the process. We consider that the first type of information is essential for scientists in the field of citizen science that do not have a background in modelling (but that can, for example, research data collection methods to address modelling needs). The second type of information is an essential component of the manuscript and, although described via text, making it explicit visually fulfils the objective of highlighting it in the paper. We are open to suggestions on how this image could be enhanced.

**Comment #6:** I suggest merging Section 1.2 - Article outline with the end of the Introduction (page 2, line 30).

**Authors' response:** Thank you for the suggestion, we will consider it. However, we will need to see how the manuscript is changed due to all the suggestions of reviewers and readers and based on that we will see if the outline and the end of the introduction can still be merged.

**Comment #7:** There are some unnecessary wording throughout the paper, for example: "We have seen in the previous section that" and "In this section we intend to" (page 14, lines 4-5).

**Authors' response:** Thank you for the suggestion, the paper will be thoroughly scanned for unnecessary wording and changed accordingly.

---

## Author Comment (AC5) · 21 Nov 2017

Thank you for the follow-up feedback. Please find below the answers to your comments.

**Comment #1:** The issue concerning the CAPTCHA plug in is not really about its definition, but about HOW it will be used as a "volunteered contribution". How, for example, random images of a deforested area chosen for security reasons will contribute to the monitoring of land use?

**Authors' response:** We will clarify the manuscript's text by saying that the process of tagging images for land use is uncorrelated to the CAPTCHA, to the test of distinguishing computers from humans. Tagging is a task performed after the test, on the same platform.

**Comment #2**: Additionally, despite the content of the outline, it makes more sense that it comes at the end of the introduction, providing readers with an initial and general idea of what will follow. The current topic 1.1 could come together with the description of data sources used and papers selection (to be included), as part of a methodology section.

**Authors' response:** Based on the suggestions, we will move the outline to the end of the introduction, keeping section 1.1 and creating a section "1.2 Review approach", which will detail the methodology taken to select the papers included in the review. We would like to reiterate here, and will strengthen this in the text of the manuscript that the intention is not that citizen science is the focus of the manuscript, but the data obtained from it, thus maintaining its discussion within the introduction section.

---

## Author Response (AR1)

**Department of Integrated Water Systems & Governance**

Thaine Herman Assumpção
t.hermanassumpcao@un-ihe.org

IHE Delft Institute for Water Education
Westvest 7
2611 AX Delft
The Netherlands

Author's response – HESS-2017-456

**Date:**
04 January 2018

Dear Editor,

Thank you for the evaluation report and for giving us the opportunity to revise the manuscript. We would like to thank the referees for taking time to review the manuscript. Their comments delivered insightful and enriching recommendations on how to improve the manuscript's content, scientific quality and readability. We also appreciated the comments made by a reader who was not a referee. A revised manuscript was prepared based on the comments we have received.

In this response letter we present the response, as follows: Section 1 contains the explanation and responses to all points raised by reviewers; Section 2 contains a list of all relevant changes made in the manuscript; Section 3 contains the revised manuscript with track changes; and Section 4 contains the revised manuscript text without track changes, for easy reading.

We hope the comments and requirements for publication are met in the revised manuscript. In case there are more concerns, please let us know for further corrections and improvements.

We are looking forward to your decision.

Yours sincerely,

Thaine Herman Assumpção and co-authors

**1. Authors' responses to reviewer's comments**

The response to all comments raised by the referees, reflected in the interactive discussion section, are presented here in detail. The page and line numbers in the authors' response refer to the revised version of the manuscript, the one that contains the marked-up changes (see Section 2 of this document).

Note: Modifications and additions to the response, as compared with the one in the interactive discussion, are highlighted in the authors' response, as underlined text.

**Anonymous Referee #1 – RC1**

We thank the reviewer for taking the time to review this paper and for providing useful feedback. Your input is valuable in improving the scientific quality of the paper and its readability. Please find below our answer addressing your comments.

**Comment #1:** *Dear editor, I went through the paper entitled "Citizen observations contributing to flood modelling: opportunities and challenges" by Assumpcao et al. Bringing people's idea and their involvement in science (citizen science) is becoming significant globally. This paper is exactly what lies behind the role of citizen science in combating the flooding by modelling. However, I find the paper is quite difficult to follow in its current form. This also has no such in-depth assessment of the role of science in mitigating climate-induced flood events/hazards.*

**Authors' response:** We acknowledge that the assessment of the role of citizen science in mitigating climate-induced flood events/hazards is not addressed in the present article and that is because the focus of the paper is different, particularly it is to review the existing scientific literature regarding the actual and potential crowdsourced data for flood modelling. From that perspective, climate-induced flood events/hazards do not bring different challenges for citizens' data collection compared to "regular" flood events. Of course citizen science is much broader than only crowdsourcing of data, but such broad perspective is outside of the scope of this article. Regarding mitigation, in the article's Introduction (page 2), we are mentioning the review of Horita et al. (2013); and the studies of Dashti et al. (2014) and Oxendine et al. (2014). They are addressing disaster management and damage data collection, including the role of citizen science for mitigation of floods in general. In the present article the analysis is made for model improvement, but the model may have multiple purposes (e.g. flood risk or ecosystem conservation). The paper determines what are the benchmarking difficulties and benefits of collecting flood-related data by citizens and of integrating them into models, for the purposes of model set up, calibration, validation, simulation and forecasting.

The improved explanation emphasizing the aim of the review is added in the Introduction section on page 3, lines 1-5.

**Comment #2:** *The synthesis/review would have been much useful and interesting if this were focused on one or two key objectives. For example, how citizen science would link to model building process based on crowdsourced data and how citizens themselves would be benefitted provided the feedbacks for the model improvement.*

**Authors' response:** The approach taken was to group and analyse the studies in which crowdsourced data was integrated into each part of the flood modelling process. We could not take a different approach because unfortunately the literature on specific parts is scarce (e.g. in Table 5, Page 19, we

found just 6 studies on the model building process). Hence, the review could not be limited to few particular aspects. Similarly, although a paper on citizen's benefits from model improvement would be useful and interesting, this is a recent topic that has not been explored enough and there are not enough publications to date so that a review is required or can be made.

**Some specific comments:**

**Page 2 Line 10-15:**

**Comment #3:** *what are the valuable contributions? elaborate*

**Authors' response:** As suggested we further elaborated, on page 2, lines 15-19, of the revised manuscript. For example, the CITI-SENSE project managed to simultaneously collect perception data and acoustic measurements in an approach that can be used to develop citizen empowerment initiatives in case of noise management (Aspuru et al., 2016).

**Page 2 Line 22-26:**

**Comment #4:** *what are three projects? provide the summary*

**Authors' response:** The manuscript was changed to include such a summary, on page 2, lines 31-32.

**Page 4 Line 19:**

**Comment #5:** *please define 'CAPTCHA plug in framework', not all readers would necessarily know about it*

**Authors' response:** A footnote was added to the manuscript in order to clarify the concept of a CAPTCHA plugin (page 6, footnote 2):

"CAPTCHA stands for 'Completely Automated Public Turing test to tell Computers and Humans Apart'. It is a test evaluating if the subject is human, which is used in websites to provide security. After the test is done the user can be asked to perform extra tasks, for example, tag images."

**Page 10 Line 12-17**

**Comment #6:** *what level of citizens will get involved to generate data globally as many citizens are devoid of IT technology?*

**Authors' response:** Iwao et al. (2006) did not provide any information on the profile of citizens, nor on engagement strategies, although the lack of data in certain regions was shortly addressed. However, as stated in the Citizen Science section of the manuscript (page 3), the review did not discuss the mechanisms of citizen engagement and participation, as this is a research topic on its own and we focus on data integration. To address this issue, also raised by a comment of a reader in the HESSD interactive discussion, explanations were added on page 4, lines 20-22.

**Page 15, Fig. 6:**

**Comment #7:** *perhaps Fig. 6 holds the core concept of the paper, where the citizen science link to modelling and its application*

**Authors' response:** Though the figure is a core concept of the paper, the paper structure is such that first the wider scope of the paper is defined, laying all the literature that has the potential to contribute to flood modelling in terms of flood-related data. This literature is characterized and analysed for advantages and disadvantages. Then, it is presented an in-depth analysis of the scientific contributions to each part of the modelling cycle. The existing literature is evaluated in terms of its information

content and analysed to check how much it matches model requirements. Finally, opportunities and challenges are identified. Following this structure, Figure 6 is presented in a later section.

**Page 18 Line 23:**

**Comment #8:** *please provide what consequences of uncertainty in data mining and how this is improved?*

**Authors' response:** The consequence of uncertainties, including the ones of data mining, is low model performance. We consider that the higher the uncertainty, the lesser the quality of the data. Hence, because data obtained through data mining has, in general, more sources of uncertainty (from value, geotagging and timestamping), they can potentially be of lesser quality and result in models with low performance. As suggested by a reader, this was further extended in the new version of the manuscript on page 21, lines 7-8; and on page 22 lines 15-16.

To date, in modelling studies, there are only few studies that quantify the uncertainty from crowdsourced data, the impact on model performance or that consider methods for its reduction. To remain neutral, we did not include in the manuscript anything beyond what is in the literature, thus we do not include a discussion on how to improve the situation in modelling.

**Anonymous Referee #2 – RC2**

We would like to thank the reviewer for the revision. We appreciate the comments provided, that deliver insightful and enriching recommendations on how to improve the content of the paper. We have addressed your comments individually in the text below.

**Comment #1:** *This paper present an interesting and fairly complete review on the use of crowdsourcing for flood modelling purposes. The effort to try and characterise the reliability and uncertainty associated to different types of data and different methods of involving citizens in collected them is worth highlighting. I would limit my review to three general comments: (1) There is no mention in the paper of the diversity of models that are used for flood modelling, and whether they are more or less suited for integrating the different types of citizen observations. Arguably, one of the challenges for hydrologists could be to design models specifically for that purpose. At least, it would have been interesting to have some information of the kind of models used in the studies analysed in the paper.*

**Authors' response:** The manuscript was modified to include an explanation on types of flood models (fluvial, pluvial, coastal and drainage) on page 17, lines 8-14. The matter of suitability is not addressed, mainly because the considered papers are not addressing the suitability. However, we found this comment very valuable and we added more information on the kind of models used in the reviewed studies (page 19, table 5).

**Comment #2:** *(2) the question of time is only very briefly discussed, while in flood modelling, and particularly for real time flood forecasting, this is an critical issue: models not only require the highest water level or the maximal flooded area extension (which are, I guess, when most of the observations are done), but high resolution data during the rising part of the hydrograph. What have been done to collect this information, and/or what type of participatory approach should be organised to do so?*

**Authors' response:** In the section 'Crowdsourced data information content' on pages 21-22, we discuss the question of time within each part of the flood modelling cycle. Flood forecasting is not included because citizens cannot provided forecasting data. We acknowledge that we do not consider

calibration and validation for specific purposes and thus do not consider them done specifically for obtaining an operational model for flood forecasting. With that in mind and in view of the reviewer's comment, we changed on page 21, Table 6, in the column 'Calibration Validation', the temporal coverage to 'Discrete/Continuous' and the spatial coverage to 'Discrete/Distributed'. A remark was added to the table mentioning that the data properties for calibration and validation depend on the purpose of the model.

Moreover, the discussion was extended to accommodate such view and answer the question on what has been done to collect this time sensitive information (page 21, lines 23-26; page 22, lines 1-3). Organisation of participatory approaches are not discussed as they are outside the scope of the proposed article.

**Comment #3:** *(3) In the same line, rainfall is almost absent in the discussion. As far as I know, crowdsourcing have also been used to obtain spatially distributed rainfall, and many extreme storm events are characterised by a high spatial variability of rainfall, so I suspect that this type of citizen observation could be useful.*

**Authors' response:** We agree that contextualization of the rainfall component is lacking and this was added to the manuscript (page 7, lines 7-11). We mentioned its importance for certain types of flooding and provided pointers to articles on crowdsourced data for rainfall. We acknowledge that citizen contributions could be useful for observation of this variable, however, we did not include rainfall in the flood-related crowdsourced data section because it was partially covered by the review of Buytaert et al. (2014) and totally covered by the review of Muller et al. (2015). Rainfall is a variable of greater importance for hydrological models, whilst the review focusses on a hydrodynamic representation of floods.

**Anonymous Referee #3 – RC3**

We thank the reviewer for providing feedback on the quality of the paper. The review is valuable for making the paper clearer and more structured and the comments are highly appreciated. Please find below our response to the provided comments.

**Comment #1:** *This paper addresses a very timely and interesting topic: citizen science and its use in flood modelling. It will provide some guidance to researchers struggling with the lack of traditional data and at the same time resistant to adhere to alternative data sources. Overall, the text is rather fluid and well written, but in topic 3, "crowded source data in flooding modeling", the explanation of some uses of citizen data in modeling is confusedly described and could benefit from a restructuring of description of uses.*

**Authors' response:** Following the reviewer's suggestion, the description of uses in Section 3 was restructured (page 19, lines 6-7; page 20, lines 1-32; page 21, lines 1-4).

**Comment #2:** *Also, despite the relatively large number of papers gathered, the revision process and papers selection is not fully described. Thus, for a synthesis paper, it will be worth proving a perspective on how exhaustive were the efforts undertaken in the collection and selection of relevant studies, and the data sources consulted.*

**Authors' response:** The manuscript was extended to inform that the papers' collection was done through multiple platforms (e.g. Scopus and Google Scholar), exemplifying used keywords (page 3, lines 7-14). Additionally, explanation on the selection criterion for consideration was given, which is

the generation/use of flood-related crowdsourced data, as well as explanation on why certain articles were not selected.

**A few minor points include:**

**Comment #3:** *In Figure 1, only level one is termed crowdsourcing, not level 2, as stated in the text (page 3, lines 30-31).*

**Authors' response:** The sentence was rephrased (page 4, lines 22-24).

**Comment #4:** *It is not clear how the CAPTCHA plug in works as a volunteered contribution; please provide a better explanation.*

**Authors' response:** Clarification regarding the CAPTCHA plug-in was done by means of a footnote, as also requested by another reviewer (page 6, footnote 2).

**Comment #5:** *Figure 2 does not seem relevant, I suggest excluding it; while Figure 6, in its present form, does not seem very informative.*

**Authors' response:** Figure 2 was included as an introductory example of framework for analysing crowdsourced data. We acknowledge that it does not attend other purposes in the previous version of the manuscript. As per suggestion of a reader that commented on HESSD interactive discussion, we included a modified version of Figure 2 further in the text, changed to include the reviewed literature (page 14, figure 3). The motivation behind increasing the relevance of such a figure is two-fold: exposition to the interested reader of classification systems of citizen science approaches; connect at a superficial level with social studies that evaluate these classifications, to increase the integration among disciplines.

Figure 6 presents visually two types of information: the components of the flood modelling process and the data necessary for each component; citizen contributions within the process. We consider that the first type of information is essential for scientists in the field of citizen science that do not have a background in modelling (but that can, for example, research data collection methods to address modelling needs). The second type of information is an essential component of the manuscript and, although described via text, making it explicit visually fulfils the objective of highlighting it in the paper. We are open to suggestions on how this image could be enhanced.

**Comment #6:** *I suggest merging Section 1.2 - Article outline with the end of the Introduction (page 2, line 30).*

**Authors' response:** In HESS interactive discussion we said we would consider this suggestion. The outline has been merged with the end of the Introduction (page 3, lines 16-22).

**Comment #7:** *There are some unnecessary wording throughout the paper, for example: "We have seen in the previous section that" and "In this section we intend to" (page 14, lines 4-5).*

**Authors' response:** Thank you for the suggestion, the paper was thoroughly scanned for unnecessary wording and changed accordingly (page 17, lines 4-7, page 18, lines 7-8).

**Anonymous Referee #3 – RC4**

Thank you for the follow-up feedback. Please find below the answers to your comments.

**Comment #1:** *The issue concerning the CAPTCHA plug in is not really about its definition, but about HOW it will be used as a "volunteered contribution". How, for example, random images of a deforested area chosen for security reasons will contribute to the monitoring of land use?*

**Authors' response:** We clarified the manuscript's text by saying that the process of tagging images for land use is uncorrelated to the CAPTCHA, to the test of distinguishing computers from humans. Tagging is a task performed after the test, on the same platform (page 6, lines 2-3).

**Comment #2:** *Additionally, despite the content of the outline, it makes more sense that it comes at the end of the introduction, providing readers with an initial and general idea of what will follow. The current topic 1.1 could come together with the description of data sources used and papers selection (to be included), as part of a methodology section.*

**Authors' response:** In HESS online discussion, we proposed to keep section 1.1 and to create a section "1.2 Review approach". Upon revision of the manuscript, we realize that the review approach could be summarized in a paragraph and that there was no need for a separate section. Thus, as mentioned in the response to the previous comment, we have added to the end of the introduction the review approach and the article outline (page 3, lines 7-22). We would like to reiterate here, and strengthened this in the text of the manuscript (page 3, lines 1-3), that the intention is not that citizen science is the focus of the manuscript, but the data obtained from it, thus maintaining its discussion within the introduction section.

**Interactive comment – SC1**

Thank you for finding the paper timely and for the appreciation of the review paper. Authors would like to thank M. Moy de Vitry for taking time to review the paper and add to the ongoing discussion. The comments and suggestions received are of high value, and based on them we made improvements to the manuscript. Please see below the answers to the comments.

**Comment #1:** *The review of how citizen observations have been used in flood modelling research is useful and very timely. The main value of the review is in mapping out the different case studies, identifying trends, and pointing out research gaps. Minor revisions are recommended:*

**Page 1 line 27:**

**Comment #2**: *Do the authors refer to the general need for data in modelling, or specifically to monitoring data used for calibrating the models?*

**Authors' response:** Authors are referring to general data needs for modelling floods, no special distinction for calibration is made. Thank you for pointing out the confusion. In order to clarify this issue to the reader an additional statement was added to the manuscript (page 1, lines 28-29).

**Comment #3**: *The example in the second sentence "This is especially true.." requires some explanation.*

**Authors' response:** More explanation was added, by rephrasing the sentence (page 1, line 29; page 2, line 1).

**Page 3 line 26:**

**Comment #4:** *Effort is made to present two classification systems. However, these classifications are not used in sections 2 and 3.*

**Authors' response:** These classifications are not introduced for the purpose of further classifying other papers, but for opening the discussion and debate on the existing reviewed literature. The first classification system (i.e. level of engagement), aims to explicitly say that discussion on advantages/disadvantages of collection/analysis methods, as well as their purposes, is strictly addressing contributions in terms of quantitative data (i.e. contributions towards flood modelling); and it does not address the advantages/disadvantages of contributions from other types of involvement. For example, it is out of the scope of the article to discuss tacit knowledge or social media mining having the (possible) disadvantage of not fostering awareness. For further clarification, the new version of the manuscript was amended (page 4, lines 20-22).

The second classification system was made to provide a reflection of such components (implicitly/explicitly geographic and implicitly/explicitly volunteered) when data is obtained from citizens. Based on this and a follow-up comment, we added a Figure where we place on the framework the studies cited in this paper; and we provided an analysis of such result (page 13, lines 12-16; page 14, figure 3).

**Page 4 line 10**

**Comment #5**: *It is unclear why geo-tagged information is not explicitly geographic.*

**Authors' response:** For clarity, in the beginning of such paragraph an explanation was added (page 5, lines 5-6).

**Page 4 lines 15-20:**

**Comment #6**: *It does not seem appropriate that SCENT is given a prominent position in this review paper, which should review published literature and not ongoing projects.*

**Authors' response:** As mentioned in the acknowledgements, this review and research related to it are supported by the H2020 project, SCENT. Therefore, it is natural that the ideas generated within the project, which aims at covering scientific gaps, are properly acknowledged in the paper text as well. The inclusion of SCENT has as objective to illustrate the classification system, taking advantage of the fact that in the project the four classes are being covered. For clarity, it was not chosen to include published literature in this part of the article without analysing it first. As per suggestion of the reviewer, we presented the same scheme later on, where such literature was included (page 14, figure 3).

**Page 5 Figure 2:**

**Comment #7**: *Fig 2 illustrates nicely how specific examples are classified within Craglia et al.'s definition, and therefore more examples would be beneficial. It would be even better if the examples were taken from literature.*

**Authors' response:** Thank you for this suggestion, we took it into account and expanded in the second version of the manuscript (page 14, figure 3).

**Comment #8:** *SCENT should be removed from the figure.*

**Authors' response:** The justification of SCENT's inclusion in the figure has been provided in a previous comment. This figure sets the scene for the second one that was added based on the reviewer's suggestion.

**Comment #9:** *it is unclear why the CAPTCHAs are neither implicit nor explicit.*

**Authors' response:** In the image CAPTCHA plugin is both implicit and explicit. The text was modified for clarification (page 6, lines 1-2).

**Page 6, lines 1-2:**

**Comment #10**: *Have studies such as Merkuryeva et al. (2015) been included in the review? please specify.*

**Authors' response:** No, they have not been included. The text was modified for clarification (page 3, lines 11-14).

**Comment #11:** *The citation is not necessary.*

**Authors' response:** We acknowledge that the citations do not serve a purpose other than being examples. However, as a review paper, we consider that different aspects of the literature should at least be exemplified, in a way that the interested reader may wish to explore topics not covered in the review.

**Page 6, line 18-20:**

**Comment #12:** *It is unclear why the text example is provided in the same paragraph as the images/videos and not in the previous paragraph*.

**Authors' response:** Thank you for bringing up this misunderstanding. The text examples are related to non-quantitative text that is converted to quantitative measures. As the section's first paragraph is about quantitative crowdsourced data and the second is about qualitative ones, this information fits better in the second paragraph. For clarity, the second paragraph was modified (page 7, line 26-31; page 8, line 1).

**Page 7, Table 1:**

**Comment #13**: *It would be good to split the column 'case study' into two columns 'location' and 'flooding type'*

**Authors' response:** The columns were split in all tables into 'Flood Type' and 'Location' and studies with flood-related crowdsourced but without floods will be classified as 'No flooding'.

**Comment #14**: *What ordering is used in the table? publication year might make sense.*

**Authors' response:** The ordering used in the table was done by grouping papers with similar measurement/analysis methods, followed by the order monitoring, mapping and modelling. This is similar to the way the analysis is done.

**Page 12, Figure 3:**

**Comment #15:** *The review extends to April 2017 - has the publication count for the year 2017 been normalized?*

**Authors' response:** No, it has not been normalized. We understand your reasoning, but our focus is on the content and interpretation, not on a precise, numerical analysis of the contributions. Thus, because of the small numbers of contributions per type of publications, for simplicity, we decide not to normalize.

**Page 13, line 7:**

**Comment #16:** *Flickr and Picasa are products, it is better to refer to photo sharing services.*

**Authors' response:** We changed to the proposed terminology.

**Comment #17:** *what is exactly meant with 'mining', and how does that entail low-quality data?*

**Authors' response:** Mining refers to the extraction of specific data from a dataset. For example, tweets can be mined from Twitter for a certain period of time and for tweets that contain the word 'flood'. We expanded the first appearance of this term to include such qualification and make it clearer (page 8, lines 3-4). Crowdsourced mined information has the possibility of not having a precise time-stamp or geotag. Thus, there is uncertainty related to it. We consider that the higher the uncertainty, the lesser the quality of the data.

**Page 18, lines 19-25:**

**Comment #18:** *The discussion on reliability and volume of data is interesting and necessary, but the statements do not seem to make good of the review that was conducted. Do none of the papers attempt to quantify uncertainty?*

**Authors' response:** Yes, some of the papers do. We expanded this discussion to include more information (page 22, lines 15-22).

**Comment #19:** *Is the volume of data per type (water level, velocity, etc.) available comparable to the number of case studies?*

**Authors' response:** We have not computed the volume of data for each data type. At the moment we estimate that they are directly proportional to the number of case studies. Unfortunately, it is not possible to get the exact number as in some cases more than one variable is collected and no distinction in the overall count is provided.

**Page 19, lines 20-26:**

**Comment #20**: *The language used is imprecise*.

**Authors' response:** The language was rephrased. See below.

**Comment #21:** *"interactions between citizen science and water resources"*

**Authors' response:** It was rephrased (page 23, lines 20-21).

**Comment #22:** "Deal with uncertainty"

**Authors' response:** It was rephrased (page 23, line 26).

**2. List of relevant changes made in the manuscript**

The relevant changes made in the manuscript are described per article section.

1. Introduction

In this section we emphasized the article's aim, included an explanatory paragraph on the review approach and removed section 1.2 by putting the article's outline at the end of the introductory text. Changes were also made to improve the clarity of some concepts.

2. Flood-related crowdsourced data

The beginning of this section was changed to discuss precipitation. The tables on the sub-sections were changed: the column 'Case Study' was split into 'Flood Type' and 'Location'. In the last sub-section, on Summary Analysis, a figure similar to Figure 2 was added, displaying the discussed studies in the framework presented in Figure 2 and analyzing the results.

3. Crowdsourced data in flood modelling

In this section, explanation on types of flood models was added, as well as information on the types of flood models used in the discussed papers. The text description of uses of crowdsourced data in the reviewed studies was restructured for clarity. Lastly, in the sub-section on crowdsourced information content, temporal dimension considerations in calibration and validation were introduced and the discussion on uncertainty and volume of crowdsourced data was expanded.

4. Opportunities and challenges

No relevant changes were made.

5. Conclusions and recommendations

No changes were made.

**3. Marked-up version of the manuscript**

This section provides the marked-up version of the manuscript. The following notation was used:

- Text that was inserted appears in red;
- Text that was deleted appear in strikethrough red;
- Black vertical track lines in the left margin indicate a change on the adjacent line.

[revised manuscript text omitted]

| | Collection | Analysis |
|---|---|---|
| Reading water level gauge | Measurement (+)
Instalation of gauges (-) | Analysis (+) |
| Collection of data from social media/Flickr/Picasa | Social media/Flickr/Picasa mining (-)
Location and time (-) | Flood extent analysis (+)
Water level and velocity analysis (-)
Field visit for water level (-) |
| Collection of data from Youtube | Youtube mining (-)
Location and time (-) | Established method (+)
Field visit (-)
Camera and other set ups (-) |
| Crowdsourced data | Measurement (+)
App/website (-) | Analysis (+) |
| Visual analysis of tagged images | | Analysis (+)
App/website (-) |
| Open Street Maps editing | Measurement (+)
OSM land cover types (-) | Analysis (+) |
| Crowdsourced video | Measurement (+)
App/website or youtube mining (-) | Established method (+) |

[revised manuscript text omitted]

---

## Referee Report (RR1)

Dear editor,

I went through the revised version of the paper entitled "Citizen observations contributing to flood modelling: opportunities and challenges" by Assumpcao et al. The manuscript has improved greatly. The area of the weak assessment that I had indicated in the original submission has now been elaborated. However, the point that I also had made on climate change, for example, how rainfall episodes would be inferred by information gathered from the citizen science would have been useful to explore. The reason is: both fluvial and pluvial floods have severe inputs on downstream landscapes that are also largely associated with upstream modifications of flow regimes due to the variation in precipitations. I understand that there may not have such publications available, but this information would be important to draw the role of citizen science in flood management aspect.

Other minor comments

Page 1 Line 20-21. Awkward sentence

Page 2 Line 7-9. Provide references

Page 3 Line 5-21. Can be the Method section

Page 3-4 Section 1.1. Can also fit in Introduction, the list of references can be summarised as short description of citizen science

Page 5 Line 9-11. Too many 'explicitly' worlds

Page 5 Line 13. 'Water level' and 'velocity' could be used with better phrases such as 'flood water level or flood inundation' and 'flow rate'

Page 11 Line 3. 'roughness' clarify further

Page 12 Line 11-12. No publications in the use of drone?

Page 16 Line 4-20. No references provided

Line 17 Fig. 7. Adopted from?

Page 19 Line 25-26. Hydrological models vs Hydrodynamic models? better clarify wherever needed

Page 21 Line 16. 'the exact number of what'?

Page 23. In the conclusions/recommendations section, it would also be worth mentioning that the studies of this kind would reciprocally benefit/aware the citizens to get involved so that the data collected would be better quality and subsequently improved simulation and modelling of floods.

---

## Author Response (AR2)

**Department of Integrated Water Systems & Governance**

Thaine Herman Assumpção
t.hermanassumpcao@un-ihe.org

IHE Delft Institute for Water Education
Westvest 7
2611 AX Delft
The Netherlands

Author's response – HESS-2017-456

**Date:**
24 January 2018

Dear Editor,

Thank you for the evaluation report. We appreciate the second round of comments from Anonymous Referee #1, which we have all addressed.

*As per request of Comment #12, we added a footnote explaining the difference between hydrological and hydrodynamic models. However, we consider that this is too basic information for the readers of HESS and we kindly ask the editor to consider removing footnote #3 on page 7 (marked-up manuscript).*

In this response letter we present the response, as follows: Section 1 contains the explanation and responses to all points raised by the reviewer; Section 2 contains a list of all relevant changes made in the manuscript; Section 3 contains the revised manuscript with track changes; and Section 4 contains the revised manuscript text without track changes, for easy reading.

In case there are more revisions needed, please let us know for further corrections and improvements.

Yours sincerely,

Thaine Herman Assumpção and co-authors

**1. Authors' responses to reviewer comments**

The response to the comments raised by the Anonymous Referee #1 are presented here in detail. The page and line numbers in the authors' response refer to the revised version of the manuscript, the one that contains the marked-up changes (see Section 3 of this document).

**Anonymous Referee #1 – RC1**

We thank the reviewer for taking the time once again to review the revised manuscript and for providing useful feedback. Your input is valuable in improving the scientific quality of the paper and its readability. Please find below our answer addressing your comments.

**Comment #1:** *Dear editor, I went through the revised version of the paper entitled "Citizen observations contributing to flood modelling: opportunities and challenges" by Assumpcao et al. The manuscript has improved greatly. The area of the weak assessment that I had indicated in the original submission has now been elaborated. However, the point that I also had made on climate change, for example, how rainfall episodes would be inferred by information gathered from the citizen science would have been useful to explore. The reason is: both fluvial and pluvial floods have severe inputs on downstream landscapes that are also largely associated with upstream modifications of flow regimes due to the variation in precipitations. I understand that there may not have such publications available, but this information would be important to draw the role of citizen science in flood management aspect.*

**Authors' response:** We agree that precipitation may be an important factor for flood modelling and that it would be useful to explore information gathered from citizens to infer it. Thus, as also recommended by Anonymous Referee #2, we expanded on the importance of rainfall for certain types of flooding and provided pointers to articles on crowdsourced data for rainfall (page 7, lines 1-5). However, as replied to Anonymous Referee #2, we did not include rainfall in the flood-related crowdsourced data section because it was partially covered by the review of Buytaert et al. (2014) and totally covered by the review of Muller et al. (2015). Rainfall is a variable of greater importance for hydrological models, whilst the review focusses on a hydrodynamic representation of floods. We acknowledge that there is an effect of climate change on hydrological and hydrodynamic processes, however, as pointed out by the referee there are no publications available on this aspect for floods, and therefore we only highlight its potential in the conclusions section (page 23, lines 11-12).

**Other minor comments:**

**Comment #2:** *Page 1 Line 20-21. Awkward sentence*

**Authors' response:** The sentence has been amended for clarity (page 1, lines 21-22).

**Comment #3:** *Page 2 Line 7-9. Provide references*

**Authors' response:** The first phrase is our interpretation of the scientific literature's state of the art. A reference for Citizen Observatory has been added (page 2, line 11).

**Comment #4:** *Page 3 Line 5-21. Can be the Method section*

**Authors' response:** The reviewed literature is relatively recent, hence a standard review process was used, which was described in one paragraph (no method section was added).

**Comment #5:** *Page 3-4 Section 1.1. Can also fit in Introduction, the list of references can be summarised as short description of citizen science*

**Authors' response:** We kept Section 1.1 as a separate section because it introduces specific concepts (frameworks for citizen science) that do not fit the introduction storyline. In our understanding, the introduction would be too long if Section 1.1 was there, with the reader losing track on the essence of the paper. However, this section is important to introduce citizen science concepts that will be used later in the manuscript and that many readers may not be familiarized with. Therefore, we decided to keep this section as Section 1.1.

**Comment #6:** *Page 5 Line 9-11. Too many 'explicitly' worlds*

**Authors' response:** Some of the 'explicitly' words were removed (page 5, lines 9-10).

**Comment #7:** *Page 5 Line 13. 'Water level' and 'velocity' could be used with better phrases such as 'flood water level or flood inundation' and 'flow rate'*

**Authors' response:** The phrasing was improved with the proposed terms (page 5, line 13).

**Comment #8:** *Page 11 Line 3. 'roughness' clarify further*

**Authors' response:** Thank you for pointing out the lack of explanation. We added a clarification (page 11, line 4).

**Comment #9:** *Page 12 Line 11-12. No publications in the use of drone?*

**Authors' response:** Not related to citizen science, to the best of the authors' knowledge. There are publications involving drones for land cover and topography. This information was added to the manuscript (page 12, lines 10-13).

**Comment #10:** *Page 16 Line 4-20. No references provided*

**Authors' response:** The descriptions provided in the mentioned lines were generated by us. The different types of flooding can be considered common knowledge. The flood modelling process can described in many ways, but we decided to introduce our own conceptualization to facilitate the understanding of the article. Therefore, no references are needed for this part.

**Comment 11:** *Line 17 Fig. 7. Adopted from?*

**Authors' response:** Figure 7 was generated by us.

**Comment 12:** *Page 19 Line 25-26. Hydrological models vs Hydrodynamic models? better clarify wherever needed*

**Authors' response:** A footnote was added in the first appearance of these terms (page 7, footnote 3). However, because HESS is a hydrological journal, we think that this footnote is not necessary.

**Comment #13:** *Page 21 Line 16. 'the exact number of what'?*

**Authors' response:** The phrase was completed (page 21, lines 10-11).

**Comment #14:** *Page 23. In the conclusions/recommendations section, it would also be worth mentioning that the studies of this kind would reciprocally benefit/aware the citizens to get involved so that the data collected would be better quality and subsequently improved simulation and modelling of floods.*

**Authors' response:** The Section was extended to include this possibility (page 23, lines 17-18).

**2. List of relevant changes made in the manuscript**

The relevant changes made in the manuscript were:

- Section 2 (Flood-related crowdsourced data): a footnote explaining the concepts of hydrological and hydrodynamic modelling was added
- Section 2.4 (Land cover/Land Use): the concept of roughness was added
- Section 2.5 (Topography): the description of the literature on drones was added
- Section 5 (Conclusions and Recommendations): a concluding remark on active participation was added

**3. Marked-up version of the manuscript**

This section provides the marked-up version of the manuscript. The following notation was used:

- Text that was inserted appears in red;
-
- Black vertical track lines in the left margin indicate a change on the adjacent line.

[revised manuscript text omitted]